# Frazzled promotes growth cone attachment at the source of a Netrin gradient in the *Drosophila* visual system

Orkun Akin*, S Lawrence Zipursky*

Department of Biological Chemistry, Howard Hughes Medical Institute, University of California, Los Angeles, Los Angeles, United States

**Abstract** Axon guidance is proposed to act through a combination of long- and short-range attractive and repulsive cues. The ligand-receptor pair, Netrin (Net) and Frazzled (Fra) (DCC, Deleted in Colorectal Cancer, in vertebrates), is recognized as the prototypical effector of chemoattraction, with roles in both long- and short-range guidance. In the *Drosophila* visual system, R8 photoreceptor growth cones were shown to require Net-Fra to reach their target, the peak of a Net gradient. Using live imaging, we show, however, that R8 growth cones reach and recognize their target without Net, Fra, or Trim9, a conserved binding partner of Fra, but do not remain attached to it. Thus, despite the graded ligand distribution along the guidance path, Net-Fra is not used for chemoattraction. Based on findings in other systems, we propose that adhesion to substrate-bound Net underlies both long- and short-range Net-Fra-dependent guidance *in vivo*, thereby eroding the distinction between them.

*For correspondence:
akin.orkun@gmail.com (OA);
lzipursky@mednet.ucla.edu (SLZ)

**Competing interests:** The authors declare that no competing interests exist.

## Introduction

Net, a secreted protein, and its receptor DCC together play a critical role in wiring the brain in both vertebrates and invertebrates. Net-DCC mediated axon guidance has been well characterized in the developing vertebrate spinal cord. In brief, Net expressed at the floor plate is proposed to diffuse to establish a decreasing ventral-to-dorsal gradient within the spinal cord (*Kennedy et al., 2006*; *Serafini et al., 1996*). This gradient, in turn, is proposed to promote the guidance of commissural neuron growth cones ventrally to the floor plate (*Fazeli et al., 1997*; *Serafini et al., 1996*). Classic *in vitro* studies using purified proteins and explant cultures also support a role for Net as a chemoattractant (*de la Torre et al., 1997*; *Kennedy et al., 1994*).

Net-DCC based axon guidance to the midline is an evolutionarily conserved mechanism in nervous system development. In *C. elegans*, loss of UNC-6 - UNC-40 (the *C. elegans* homologs of Net-DCC) signaling leads to dorsal displacement of axon tracts that are positioned ventrally in wild-type (*Chan et al., 1996*; *Hedgecock et al., 1990*; *Ishii et al., 1992*). And in *Drosophila*, *Net* and *Frazzled* (*Fra*, the Drosophila homolog of DCC) are required for axons to cross the midline of the embryonic ventral nerve cord (*Harris et al., 1996*; *Kolodziej et al., 1996*). Here, a number of neurons from each side of the midline send their axons contralaterally, creating the commissures of the ventral nerve cord. Net is expressed at the midline by glia and Net protein is found as a gradient that peaks at the midline. In *Net* or *fra* mutants, the commissures of the ventral nerve cord are missing or severely reduced, consistent with a loss of chemoattraction to the ligand source.

The spatial relationship between a Net-responsive growth cone and the source of Net in the fly visual system is similar to both the fly midline and vertebrate spinal cord. The fly visual system is modular, comprising some 750 columns. Different neuronal cell types, including R8, extend axons within each column, where they terminate in different layers. R8 growth cones reach their target, the

**eLife digest** The brain of the fruit fly contains hundreds of thousands of neurons, while the human brain contains more than 80 billion. Each of these consists of a cell body that bears an array of branches called dendrites, plus a single cable-like axon. During development, the neurons organize themselves into complex networks by forming connections with one another via their axons and dendrites. But it is not clear exactly how the correct connections form in the correct places.

As they grow out, axons rely on specialized moving structures at their tips – known as growth cones – to probe their environment in search of attractive and repulsive chemical signals released by other cells. When sensors on the surface of growth cones detect a target signal, they initiate processes that cause the growth cone to expand or collapse. This enables the axons to move towards or away from the signal, as appropriate. In all animals studied, proteins called DCC and Netrin form one of the best-known sensor-signal pairs. Growth cones bearing DCC sensors are thought to detect 'wafting plumes' or gradients of Netrin and then grow towards the Netrin source.

However, nobody had directly watched neurons respond to Netrin in a living intact animal. Using a type of microscope that can look deep into the developing fly brain, Akin and Zipursky have now followed the movement of growth cones on cells called R8 neurons in fruit fly pupae. Unexpectedly, Akin and Zipursky found that the growth cones of mutant flies that lack Netrin or Frazzled (the fruit fly version of DCC) navigate successfully to their intended destinations. Once there, however, the mutant growth cones were unable to attach to their targets.

Akin and Zipursky's work is consistent with other observations in a number of animal and insect systems that suggest that Netrin may not attract growth cones via wafting plumes of signal. Instead, Netrin may form a sticky trail that helps growth cones to gain traction as they crawl towards or stick to their destinations. Further experiments are now needed to test whether other neurons in fruit flies and in different animals use Netrin in this way.

M3 layer of the medulla neuropil, in two steps (*Ting et al., 2005*) (*Figure 1a*). They 'park' at a temporary target at the outer surface of the medulla, referred to as M0, and then, after a delay, extend to and terminate within the M3 layer. Salecker and colleagues demonstrated that R8 targeting to M3 requires *Net-Fra* signaling (*Timofeev et al., 2012*). R8s express *Fra; Net* is expressed by L3 growth cones within the M3 layer and is seen as a shallow gradient that stretches from M0 to M3, peaking sharply at M3 (*Timofeev et al., 2012*). R8 growth cones extend from M0 to M3 within a dense neuropil containing the processes of many different cell types. Without *Fra* or *Net*, many R8 terminals remain in M0 while others are stranded between M0 and M3 (*Timofeev et al., 2012*).

The findings in the R8 system are consistent with Net acting as a chemoattractant, a secreted ligand diffusing to form a gradient, which promotes extension of growth cones to the source of the ligand. However, a different interpretation was favored, one that proposed a local function for Net-DCC at M3 in target layer recognition (*Timofeev et al., 2012*). Central to this conclusion was the observation that a membrane-tethered variant of Net, expressed from the endogenous locus, can support wild-type targeting. The same molecular strategy had been used in the *Drosophila* embryonic midline to establish that diffusible Net is not required for the *Fra*-dependent guidance of commissural axons (*Brankatschk and Dickson, 2006*). This seminal work solidified the short-range paradigm of Net-DCC mediated axon guidance in which a soluble gradient of the ligand is not required for function.

Notably, at both the embryonic midline and visual system, tethered Net is expressed in a graded fashion comparable to the secreted form. At the midline, the boundaries of Net expression largely reflect the extended lateral morphology of cells expressing Net, the midline glia (*Brankatschk and Dickson, 2006*). Similarly, although Net is prominently expressed in L3 growth cones within the M3 layer, it is also visible as a gradient along the R8 targeting path between M0 and M3 (see *Figure 1b*). Thus, while these studies are consistent with a Net requirement that is limited to the extent of the source cells, it is not known whether the observed graded distributions are important for axon guidance to the target—the peak of the Net signal.

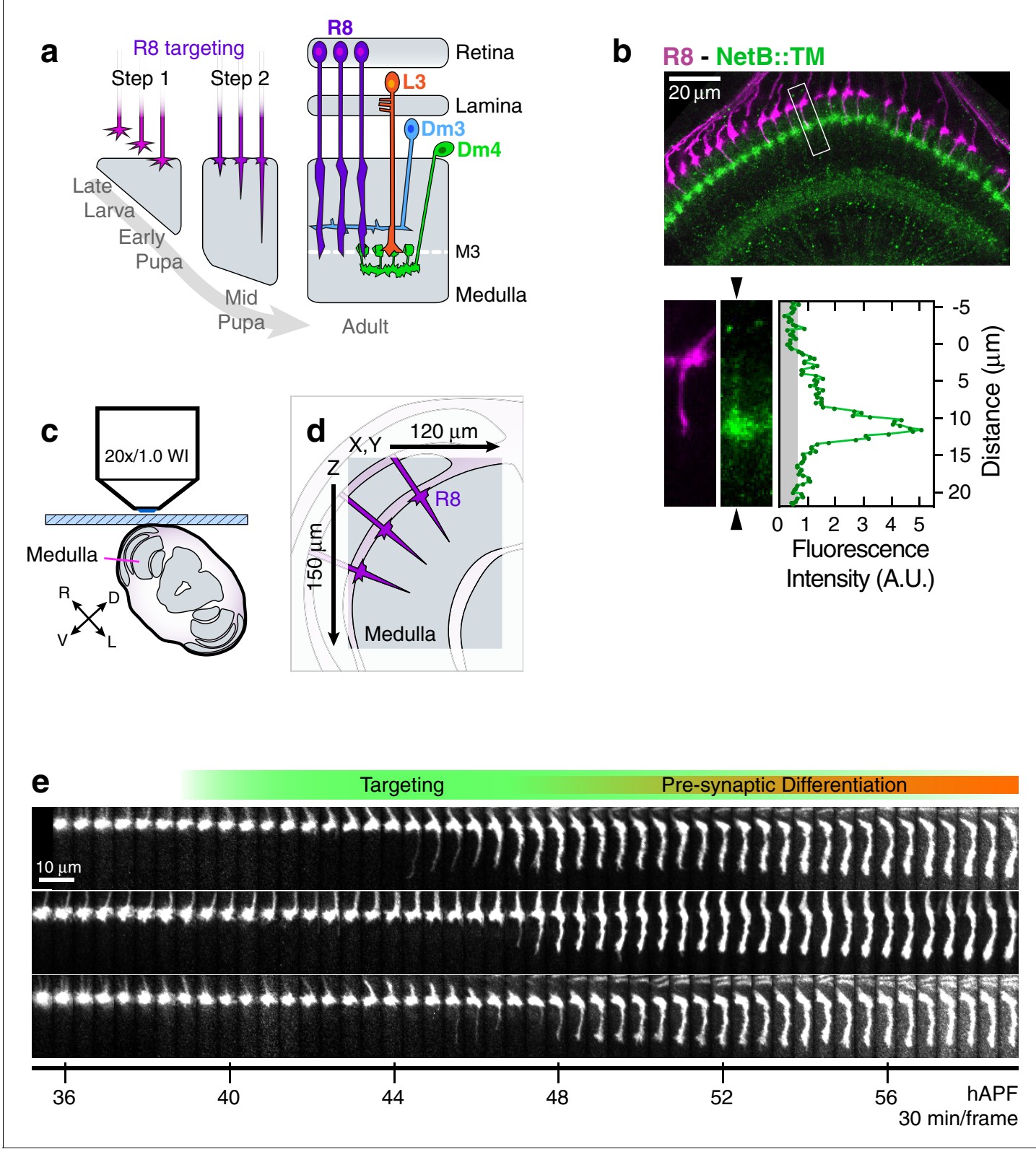

**Figure 1.** Live imaging of R8 growth cones in the developing fly brain. (a) Schematic of R8 targeting. (b), *Top panel:* Confocal micrograph of the medulla at 45 hAPF in a fly expressing a membrane-tethered variant of NetB (NetB::TM) from the NetB genomic locus. R8s (magenta, myr::tdTOM) and NetB (green, myc) are shown. *Bottom panels:* Higher magnification view of the boxed region in the top panel; the R8 and NetB::TM channels are displayed separately. Arrowheads bracketing the NetB::TM image mark the position of the linescan plotted to the right. *Graph:* Linescan of the

*Figure 1 continued on next page*

*Figure 1 continued*

fluorescence intensity in NetB::TM channel. Gray region marks background values. (**c**) Sample setup. Brain is shown in coronal section, viewed head-on. Body axes, (D)orsal-(V)entral and (R)ight-(L)eft, are marked. (**d**) Detail from (b) illustrates the imaged volume. (**e**) Three growth cones from the same WT brain. Panels were individually contrast enhanced to reveal dimmer features. See *Figure 1—figure supplement 1* for a description of the image processing work-flow.

The following source data and figure supplements are available for figure 1:

**Source data 1.** Contains numerical data plotted in *Figure 1b*.

**Figure supplement 1.** Processing 2-photon time series.

**Figure supplement 1—source data 1.** Contains numerical data plotted in *Figure 1—figure supplement 1b*.

Given the penetrance and expressivity of the *fra* and *Net* phenotypes in R8 (see below) and the genetic tools available in *Drosophila* to explore mechanisms of axon guidance, we used live imaging to explore R8 targeting to M3 in more detail. Here we report, through detailed quantitative analysis of hundreds of mutant and wild type growth cones in intact developing animals that R8 growth cones in *Net* mutants or R8 growth cones lacking *Fra* target from M0 to M3 in a fashion indistinguishable from wild type. That is, Net does not act as a chemoattractant nor does *Fra* act as a chemoattractant receptor. In addition, we present evidence suggesting that R8 growth cones can recognize the target layer without *Net-Fra*. Instead, Netrin, DCC, and TRIM9, a signaling component directly downstream from DCC (*Alexander et al., 2010*; *Hao et al., 2010*; *Morikawa et al., 2011*; *Song et al., 2011*; *Winkle et al., 2016, 2014*), are essential for attachment of a single leading process extended from R8 growth cones to the target layer. We propose that R8 growth cones reach and recognize the target layer independent of Fra, adhere to the target layer in a Fra-dependent step, and this adhesion is consolidated by a TRIM9-dependent step. These findings favor the notion that in *Drosophila Net* mediates adhesion to neuronal processes or the extracellular matrix (ECM) at the target layer rather than promoting directed outgrowth to or recognition of the target layer.

## Results

### Live imaging reveals that R8 targeting occurs via discrete steps

To compare wild-type and mutant R8 targeting, we devised a live imaging protocol to follow growth cones in intact pupae as they extend from M0 to M3 (*Figure 1c–e*, *Figure 1—figure supplement 1*, *Video 1*). This system is similar to one developed by Hiesinger and colleagues to study the cellular mechanism of neural superposition, the choreographed re-distribution of R1-6 growth cones from ommaditial bundles to lamina cartridges (*Langen et al., 2015*). In the medulla neuropil, Hiesinger and colleagues used *ex vivo* live imaging to study R7 targeting, specifically characterizing the role of the Ca2+-dependent cell adhesion molecule, N-cadherin, in mediating adhesive interaction between growth cones and the developing neuropil (*Özel et al., 2015*).

Through analysis of hundreds of wild type R8 growth cones, we characterize four distinct steps of targeting (*Figure 2a*, *Video 2*). The first, *extension*, begins with the polarization of growth cones in M0; a single thin process appears at the medial side of each growth cone (~38 hAPF). For many growth cones, this process probes into the medulla with multiple excursions and retractions, at rates reaching 1 μm/min (*Figure 2c*). *Extension* ends at 46.5 ± 1.5 hAPF with *stabilization*,

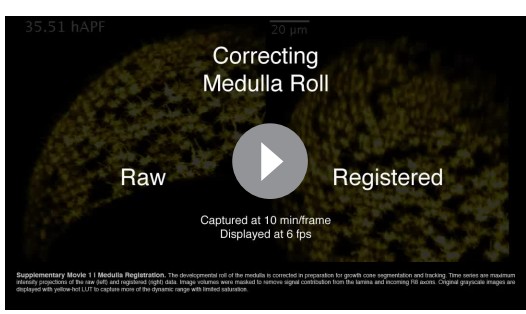

**Video 1.** Medulla Registration. The developmental roll of the medulla is corrected in preparation for growth cone segmentation and tracking.

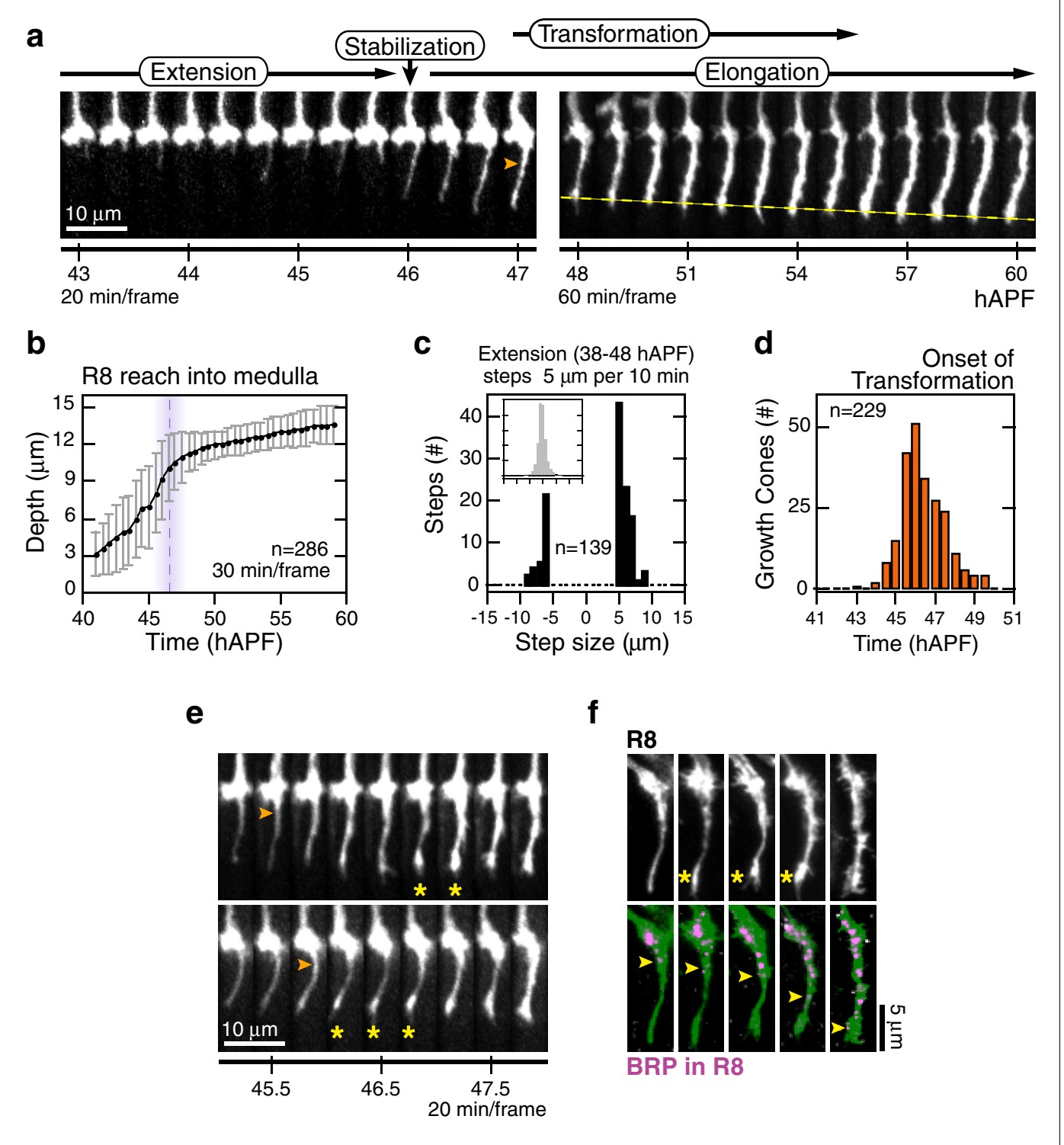

**Figure 2.** Wild-type R8 targeting. (a) Steps of WT targeting. Orange arrowhead marks the onset of *transformation*. Dashed yellow line marks R8 depth through *elongation*. See also *Figure 3—figure supplement 1b* for an illustration of transient excursions from the target layer after *stabilization*. (b) Average reach of the R8 tip into the medulla (2 animals). Error bars are standard deviation. Dashed magenta line and band mark *stabilization*. (c) Counts of frame-to-frame (Δt = 10 min) tip movements equal to or greater than ±5 μm during *extension* (3 animals). Inset is the full distribution of steps, the tails of which are plotted in the parent graph. (d) Onset of *transformation* (3 animals). (e) *Transformation* proceeds from both ends. Orange arrowheads mark the first frames in which the proximal length of the thin process begins to expand. Yellow asterisks mark the expansion of the tip; see also (f). (f) Brp, a marker for presynaptic differentiation, accumulation follows anterograde expansion (yellow arrows) during *transformation*. Panels show

*Figure 2 continued on next page*

*Figure 2 continued*

confocal images taken at 47–49 hAPF. R8s are labeled with myr::GFP. R8s express V5-tagged Brp using the STaR system (*Chen et al., 2014*). Overlay of the Brp channel with a mask of the GFP channel highlights R8-localized puncta in magenta.

The following source data is available for figure 2:

**Source data 1.** Contains numerical data plotted in *Figure 2b,c,d*.

as the tips of the R8 processes settle at 10.5 ± 1.5 µm from M0 (*Figure 2a,b*). In the third step, *elongation*, the tips of R8 projections continue to move away from M0 at the much slower rate of ~2.2 µm/hr (*Figure 2a,b*).

*Elongation* may represent the final step of active growth toward the target layer. Alternatively, R8s could be increasing their lengths in concert with the growing medulla. To distinguish between these possibilities, we compared the reach of R8 projections to a fiducial marker for an outer medulla layer, the Dm3 neurons (*Figure 1a*). In the adult, Dm3 processes project across columns to weave a meshwork at the M2/M3 layer boundary, a position that corresponds to 67 ± 5% of the depth of R8 projections (*Figure 3a*). Between 45 and 60 hAPF, the Dm3-defined layer is gradually displaced from M0 at an average rate of 0.9 µm/hr (*Video 3*). Since the projections of these neurons extend orthogonally to medulla columns, the measured movement captures the expansion of the neuropil. During *elongation*, the ratio of the distance between M0 and the Dm3 layer and the reach of individual R8 projections is 61 ± 4% (*Figure 3d* and *Figure 3—figure supplement 1*), suggesting that the R8s are increasing in length in concert with the growing neuropil. This result indicates that the target layer is reached at *stabilization*. It is possible that R8 axons actively and independently *elongate* to match the growth of the tissue. However, both *elongation* and medulla expansion continue through pre-synaptic differentiation (*Chen et al., 2014* and below), arguing that the concerted growth is supported by attachments to neighboring cells and the target layer. We conclude that R8 projections recognize and become attached to their targets before *elongation*, at the *stabilization* step, and are stretched in length as the target layer moves away from M0.

The first three steps of targeting are defined by the position and dynamics of the R8 tip; the fourth step, *transformation*, is a morphological change that begins after *stabilization* and overlaps with *elongation* (*Figure 2a,d*). At the onset of *transformation*, the original growth cone volume at M0 shrinks while the thin process fills out. The progress of *transformation* is not purely anterograde. In many cases, the apparent flow of material from the growth cone into the proximal length of the process and the expansion at the distal tip are distinguishable (*Figure 2e*). In the following ~5 hr, the R8 terminal takes on its mature form, which will eventually contain ~50 en passant synapses (*Chen et al., 2014*; *Takemura et al., 2013*). To ask how *transformation* relates to pre-synaptic differentiation, we studied the distribution of Brp in R8 projections using the STaR system (*Chen et al., 2014*), which enables cell-type specific tagging of proteins expressed at endogenous levels. We found that this marker for pre-synaptic active zones begins to populate the length of the R8 projection with the onset of *transformation*, following the anterograde expansion to eventually reach the expanded tip at the target layer (*Figure 2f*). Thus, while *transformation* is linked to pre-synaptic differentiation, the expansion of the R8 tip at the target layer is an earlier and distinct sub-step.

## Re-visiting the *Net-fra* phenotypes

One potential challenge to examining the role of *Net-Fra* in R8 targeting is the discrepancy between *fra*$^{null}$(*fra*$^3$ [*Kolodziej et al., 1996*], referred to as *fra*$^{null}$ in the main text) and *Net* phenotypes reported by Salecker and

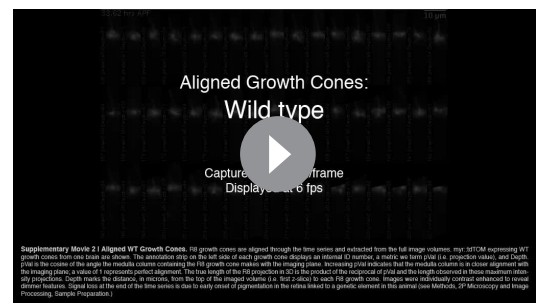

**Video 2.** Aligned WT Growth Cones. R8 growth cones are aligned through the time series and extracted from the full image volumes. myr::tdTOM expressing WT growth cones from one brain are shown.

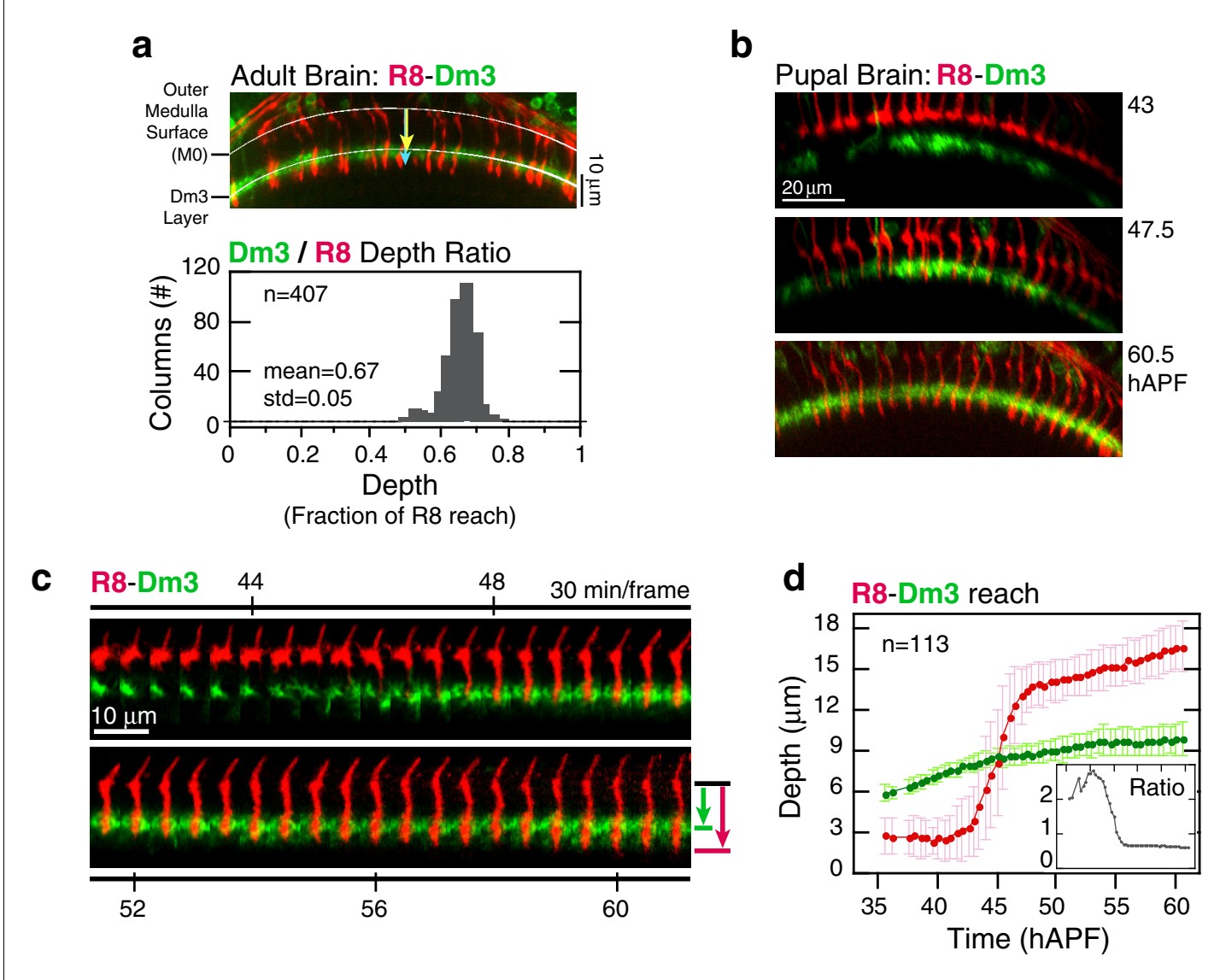

**Figure 3.** Analysis of *elongation*. (a) *Panel:* Confocal micrograph of the outer medulla in the adult brain. R8s (red, myr::tdTOM) and Dm3s (green, myr::GFP) are shown. White arcs are fits to M0 and to the Dm3 processes. Arrows, yellow and blue, mark the distance from M0 to the Dm3 layer and to the R8 tips. *Graph:* Ratio of Dm3 and R8 depths in the brain from panel (n = 407). (b) Live imaging of R8 and Dm3. Panels from live imaging of the adult brain in (a). The view presented, matching the confocal micrograph in (a), was generated post-processing. Panels were individually contrast enhanced to reveal dimmer features. Note that Dm3 processes complete their expansion into layer M2-3 during the window of observation. Despite the incomplete coverage at early time-points, the representation of Dm3 throughout the time-series is sufficient to calculate a surface fit to this fiducial layer marker (see Materials and methods). (c) Time series of an R8 and underlying Dm3 process, a fiducial marker for the M2/M3 boundary. Red and green arrows illustrate the measurements plotted in (d). (d) Average reach of the R8 tip (red arrow in (c)) and the Dm3 distance from M0 (green arrow in (c)), measured in one brain. Error bars are standard deviation. *Inset:* Ratio of the Dm3 distance to M0 and the R8 tip reach. Mean ratio between 50–60 hAPF is 0.65 ± 0.06.

The following source data and figure supplements are available for figure 3:

**Source data 1.** Contains numerical data plotted in *Figure 3a,d*.

**Figure supplement 1.** Analysis of elongation.

**Figure supplement 1—source data 1.** Contains numerical data plotted in *Figure 3—figure supplement 1a,b*.

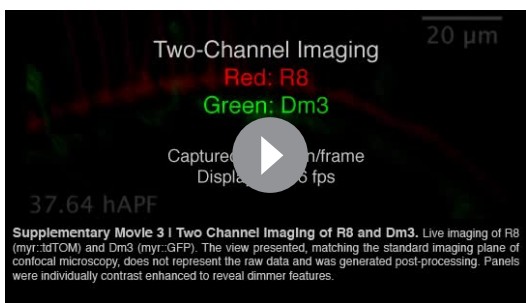

**Video 3.** Two Channel Imaging of R8 and Dm3. Live imaging of R8 (myr::tdTOM) and Dm3 (myr::GFP).

colleagues. In genetically mosaic animals where *fra* is removed specifically from R8s, ~90% fail to reach the M3 layer (*Timofeev et al., 2012*). In contrast, the failure rate in whole animal *Net* mutants (*NetA* and *NetB* double mutant, referred to as *Net* null in the main text) (*Newquist et al., 2013*) is ~50% (*Timofeev et al., 2012*). These data raise the possibility of a second Fra ligand acting in parallel with *Net*. To clarify the interpretation of the live imaging data in the context of this ambiguity, we re-visited the mutant analysis through quantitative comparison of adult phenotypes.

In the adult brain, wild-type R8 projections reach 16.9 ± 1.7 μm into the medulla (*Figure 4a*, top graph). The spread in this distribution is partly due to the 3D organization of this tissue; the absolute distance between layers changes along the two principal axes of the neuropil. Wild-type R8 projections, when expressed as a fraction of the distance between M0 and M6, reveal a more narrowly defined (0.75 ± 0.05) target layer (*Figure 4a*, bottom graph). In addition to reducing variation within one brain, this normalization approach also enables more reliable quantitative comparisons to be made between R8s from different animals and different genetic backgrounds. For *fra*^null^ R8 terminals, the normalized depth is well-described as a mixture of three distributions (*Figure 4b* and see Materials and methods). All three components fall short of the target layer. Thus, removing Fra from R8s yields a phenotype with nearly full penetrance (i.e. the number of R8s affected) and variable expressivity (i.e. the severity of the R8 phenotype).

Visual inspection of R8s in *Net* null animals suggests that a large fraction of R8s project to the M3 layer (*Figure 4d*, panel). However, the normalized depth for R8s in this genetic background can also be described as a mixture of three distributions, and the deepest reaching component (*Figure 4d*, graph and table) is nearly identical to that found in *fra* R8s (*Figure 4b*, graph and table) in terms of mean and standard deviation. In other words, the R8 depth distribution in the *Net* null background is well-described without a wild-type component (*Figure 4a*, bottom graph). To directly measure the distance between the R8 tips and the M3 layer in *Net* null animals, we used a *Net*-insensitive (*Figure 4g*) M3 marker, the Dm4 neuron (*Figure 1a* and *Figure 4h*). In wildtype, 95% of R8 tips reach past the Dm4 midline (*Figure 4i*). In contrast, only 19% of R8s in *Net* null animals extend this far. The majority of this 19% would belong to a sub-population distinct from wild type (i.e. tail of a mutant distribution); a much smaller fraction would be drawn from a purely wild-type component.

We conclude that the penetrance of the *Net* mutation for R8 targeting is much higher than originally reported and that the difference between *fra*^null^ and *Net* R8 targeting defects is one of expressivity.

To study the *fra-Net* genetic interaction, we performed epistasis analysis by reducing Fra levels in R8 cells with RNAi in the *Net* null background (*Figure 4e*). The phenotype of the double mutant was indistinguishable from *Net* null (*Figure 4f*). This result indicates that *Fra* and *Net,* as expected from a receptor-ligand pair, act in the same genetic pathway for R8 targeting. It also rules out the possibility of a second Fra ligand, acting in parallel with Net. However, the epistasis analysis does not rule out possible cell-autonomous contribution of other Net receptors. Indeed, the activity of such receptors in R8s would be consistent with the expressivity difference observed between the *Net* and *fra*^null^ genetic backgrounds.

We assessed this possibility for Dscam1 and UNC-5 with genetic mosaic analysis and did not find any obvious defects in R8 targeting. These results are consistent with the findings of a recent a cell-type specific RNA sequencing study (*Tan et al., 2015*), which showed that, at the onset of targeting, Fra is the only Net receptor expressed in R8 above 1–2 RPKM, the common threshold for noise in these types of studies.

Together, the epistasis analysis and the lack of evidence for the R8-specific activity of other Net receptors indicate that the difference in expressivity between the *fra*^null^ and *Net* phenotypes is due to a non-cell autonomous, pleiotropic effect of the *Net* mutation. Consistent with this finding, we found two other cell types with altered morphologies in the *Net* background (*Figure 4j*).

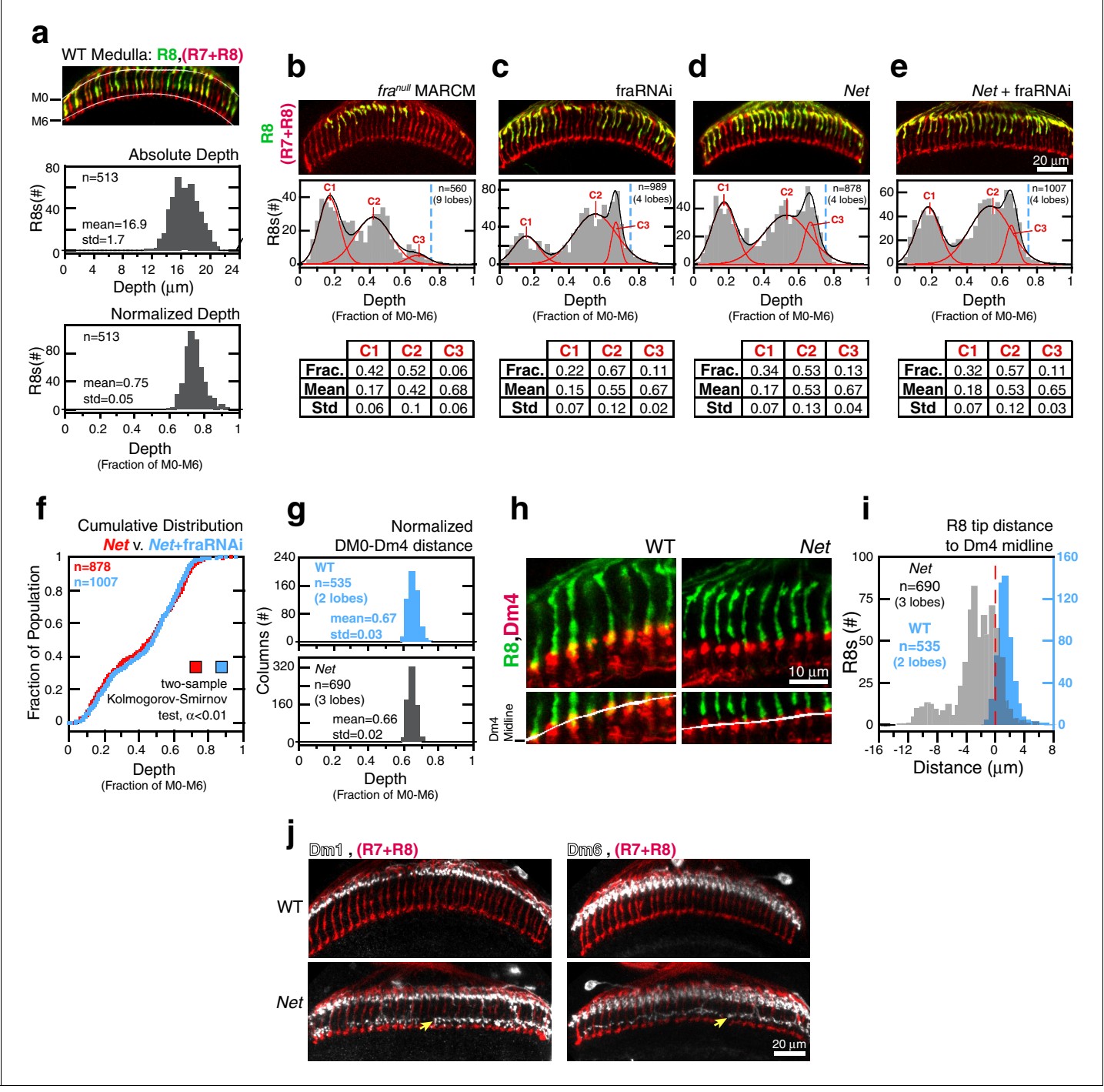

**Figure 4.** *Fra* and *Net* are in the same genetic pathway for R8 targeting. (a) *Panel:* Confocal micrograph of outer medulla. R8s (green, ~70% expressing myr::GFP) and all R cells (red, labeled with Mab24B10) are shown. *Graphs:* Absolute (top) and normalized (bottom) distance between M0 and the R8 tips measured in medulla. (b) *Panel:* Confocal micrograph of outer medulla in a MARCM brain. *fra*$^{null}$ R8s (green, expressing UtrnCH::GFP) and all R cells (red, labeled with Mab24B10) are shown. Projection depths of *fra*$^{null}$ R8s were measured using a membrane-targeted marker (myr::tdTOM, not shown). *Graph:* Normalized *fra*$^{null}$ R8s projection depths. Components (C1-3) for the Gaussian Mixture Model (GMM) fit to the distribution are plotted in red; black trace is the sum of the components. Dashed blue line marks the mean of the WT distribution from (a). *Table:* GMM parameters. Frac.: fractional contribution of each component to the fit. (c) *Panel:* Adult brain in which *fra* expression was knocked down in R8s using a cell-type specific driver (i.e. fraRNAi). Cell labeling as in (a). *Graph:* Normalized projection depths for fraRNAi R8s. *Table:* GMM parameters. (d) *Panel:* *Net* adult brain. Cell labeling as in (a). *Graph:* Normalized R8 projection depths in *Net* animals. *Table:* GMM parameters. (e) *Panel:* *Net* adult brain in which *fra* expression was knocked down in R8s using a cell-type specific driver (i.e. *fra* RNAi). Cell labeling as in (a). *Graph:* Normalized R8 projection depths in

*Figure 4 continued on next page*

*Figure 4 continued*

the *Net+fra* RNAi background. *Table:* GMM parameters. (f) Cumulative distribution of data in (d) and (e). The distributions are not distinguishable by the two-sample Kolmogorov-Smirnov test at significance level $\alpha < 0.01$. (g) Comparison of the normalized distance from M0 to the midline of the Dm4 processes near M3 in WT (top, blue) and *Net* (bottom, gray) male adult brains. (h) Confocal micrographs of the outer medulla in WT (left) and *Net* (right) adult brains. R8s (green, myr::GFP) and Dm4s (red, myr::tdTOM) are shown. White curves are fits to the Dm4 midline. (i) Absolute distance between R8 tips and the Dm4 midline (dashed red line at 0) measured in WT (blue) and *Net* (gray) adult brains. Positive values indicate that the R8 tip is past the Dm4 midline. (j) Two additional cell types with altered morphologies in the *Net* background. Dm1 and Dm6 are both multi-columnar amacrine cells with processes at M1 in the WT (top row). Both cell types generate extra arborizations at M4-M5 (bottom row, yellow arrows) in *Net* animals. Dm1s and Dm6s expressing myr::tdTOM (white) were visualized with immunohistochemistry. Mab24B10 was used to stain for all photoreceptors (red).
The following source data is available for figure 4:

**Source data 1.** Contains numerical data plotted *Figure 4a,b,c,d,e,f,g,i*

Why is the expressivity of the *Net* whole animal phenotype on R8 growth cones less severe than the removal of *Fra* selectively from R8s in mosaic animals? What is surprising is not that they are different, but rather that the pleiotropy is associated with weaker R8 expressivity. Removing *Net* from the whole animal alters the development of multiple neuronal classes in the medulla. Some, like R8, may be affected through the *Fra* pathway, while loss of Net signaling through the UNC-5 receptor may be important in other cases. We suspect that some aspect of the micro-environment in which R8 growth cones develop becomes more permissive to the progress of *transformation* in the absence of *Net* (see below). Regardless of the specific causes of pleitropy, the altered medulla environment in *Net* animals confounds mechanistic interpretation of R8 growth cone defects.

For a more refined approach to studying the role of *Net* in R8 targeting, we pursued two strategies involving genetic manipulation of the L3 lamina monopolar neuron, the principal source of Net in the target layer (*Pecot et al., 2014*; *Timofeev et al., 2012*). First, we ablated L3s through cell-type specific RNAi-mediated knock-down of a neurotrophic receptor. This experiment was originally published by our group where we showed that removal of all L3s led to a loss of Net from the M3 layer (*Pecot et al., 2014*). To compare the development of R8s with and without a home-column L3 in the same brain, we reduced the efficiency of the knock-down to achieve a partial ablation of the L3 array (~80% loss with ~20% surviving through eclosion). In this setup, all R8s with an L3 in the home column target as wild type. With R8s lacking a home column L3, we did observe targeting defects, but the phenotype was too variable to discern general patterns of growth cone behavior and draw mechanistic conclusions. We suspect that ablating L3, a major resident of the M3 layer, is too blunt a perturbation with pleiotropic consequences that go beyond removing a single secreted ligand.

As a second approach, we followed the targeting of wild-type R8s in columns with *Net* null L3s in genetic mosaics (i.e. MARCM) using live imaging. We did not observe any overt R8 phenotypes in this genetic background. The principal caveat here is that with MARCM in the lamina, mutant neurons appear as singlets or in small patches in an array of wild-type cells. We could not achieve mutant L3 patches large enough to ensure that any R8 sharing a column with a *Net* null L3 did not also neighbor at least one wildtype L3. Indeed, at 40 hAPF, L3 growth cones are very large along the dorso-ventral axis and extend into adjacent columns. Thus, it is likely that Net is contributed from wild-type L3 growth cones in neighboring columns.

In summary, the quantitative re-assessment of the mutant phenotypes and the epistasis analysis establishes that, for R8 targeting, removing either *Net* or *Fra* are equivalent perturbations. Thus, while it has not been possible to directly study the cell-type specific effect of removing *Net* on R8 targeting, we sidestepped the complications of *Net* genetics by focusing our efforts to characterize the role of Net-Fra signaling on comparing the dynamics of wild-type and *fra*^null^-mutant growth cones in genetically mosaic animals.

## Net-Fra signaling is not required for chemoattraction but promotes adhesion to the target layer

To study the role of *Fra* in R8 targeting, we combined live imaging with MARCM, a genetic strategy that relies on mitotic recombination to generate positively marked mutant cells in otherwise wild

type tissue (*Lee and Luo, 1999*). In this experimental design, we also incorporated a direct labeling scheme, to mark all R8s independent of genotype. This enabled us to compare *fra^null* and wild-type R8 growth cones projecting into the same wild type medulla (*Figure 5* and *Video 4*). Due to variations in the M0-to-M3 distance between different columns in the same wild type animal and between different samples, this internally controlled setup was essential for the detailed comparisons between wild type and mutant growth cones described below.

R8 growth cones reach the target layer without *fra* in a fashion indistinguishable from wild type. *Extension* is unaffected in mutant growth cones; a single thin process appears on the medial side and reaches into the medulla as in wild type (*Figure 5a,b,d* and *Figure 5—figure supplement 1a, b*). At ~48 hr, the tip of the thin process from these growth cones stalls at ~10 μm from M0 (*Figure 5a,b*), the approximate position of the target layer recognized by wild-type growth cones. At this point, *fra^null* development diverges from wild type in two respects. First, the distal tips of *fra^null* processes do not expand (*Figure 5a,b,g* and *Figure 5—figure supplement 2*). That is, while subtle dilations or other distinct structures may appear at this active site, these are invariably transient and are remodeled within 20–30 min to restore the thin morphology of the targeting process. Second, mutant processes do not stably adhere to the target layer. Instead, for the next ~10 hr (*Figure 5—figure supplement 1e*), as wild-type R8s passively elongate (*Figure 5h*), *fra^null* R8s actively follow a slowly advancing front in the medulla (*Figure 5g,i* and *Figure 5—figure supplement 2*), before they ultimately retract. That is, mutant R8s continue to extend and retract their thin processes, as their reach into the medulla increases over time. We term this behavior *tracking*.

We sought to assess whether *fra^null* growth cones track the moving target layer. For each wild type and *fra^null* growth cone in mosaic brains, we calculated the trendlines followed by the advancing R8 projections during *elongation* and *tracking* (see Materials and methods). Greater than 95% of *fra^null* R8s yielded *tracking* trendlines that are consistent with target layer trendlines of their wild-type counterparts (*Figure 5j* and *Figure 5—figure supplement 1c,d*). Thus, *fra^null* R8s track the target layer. This indicates that mutant R8 growth cones recognize one or more determinant of target layer specificity. The observation of *tracking* uncouples target layer recognition and attachment. In R8s, target layer recognition does not require the *Net-Fra* pathway.

The onset of *transformation* occurs on schedule in *fra^null* mutants (*Figure 5e*); however, in the absence of tip expansion, the progress of this morphological change is now only anterograde. As in wild type, Brp puncta enter mutant R8 projections following this anterograde expansion (*Figure 5f*). The progress of *transformation* is slow and fails to reach the target layer in the vast majority of *fra^null* growth cones before the final retraction after *tracking* (*Video 5*). That is, while synaptogenesis is affected in *fra^null* mutants, this is a downstream consequence of the earliest observable phenotypes—lack of tip expansion and stable attachment. The penetrance of the loss of target layer adhesion in *fra^null* growth cones is complete, consistent with our quantitative analysis of the adult phenotype. Live imaging of R8 targeting in *Net* null animals revealed similar behavior (*Figure 6*). In summary, the *Fra-Net* pathway is required for the attachment of the R8 terminal to M3 (Figure 8a). The marked expansion at the tip in wild type is consistent with Net-Fra signaling coupling a cytoskeletal response to substrate adhesion.

## Trim9 is required to consolidate tip expansion

We sought to further test whether the Net-Fra pathway mediates adhesion. To do this, we assessed the function of another gene in the Net-DCC pathway that has cell-autonomous function. Trim9 has been identified as a component of Net-DCC signaling in neural development in both vertebrates and invertebrates (*Alexander et al., 2010*; *Hao et al., 2010*; *Morikawa et al., 2011*; *Song et al., 2011*; *Winkle et al., 2016*, *2014*). Trim9 interacts through its SPRY domain with the Fra C-terminal cytoplasmic tail (*Morikawa et al., 2011*).

*Trim9* is required in R8 targeting, as assessed in fixed preparations in the adult. The vast majority of *Trim9^null* (*Trim9^91* [*Morikawa et al., 2011*], referred to as *Trim9^null* in the main text) R8s analyzed in genetically mosaic animals fall short of the target layer (*Figure 7b*). The small (~1%) fraction of mutant R8s that do overlap the wild-type depth distribution (*Figure 7a*) may represent a sub-population that targets as wild-type, or these may be the outliers of the mutant distribution. We conclude that, like *fra^null*, the *Trim9* targeting phenotype exhibits near-complete penetrance. To explore the genetic interaction between *Trim9* and *Fra*, we performed epistasis analysis by reducing Fra levels in R8 cells with RNAi in the *Trim9* MARCM setup (*Figure 7c,d*). The phenotypes of double mutant and

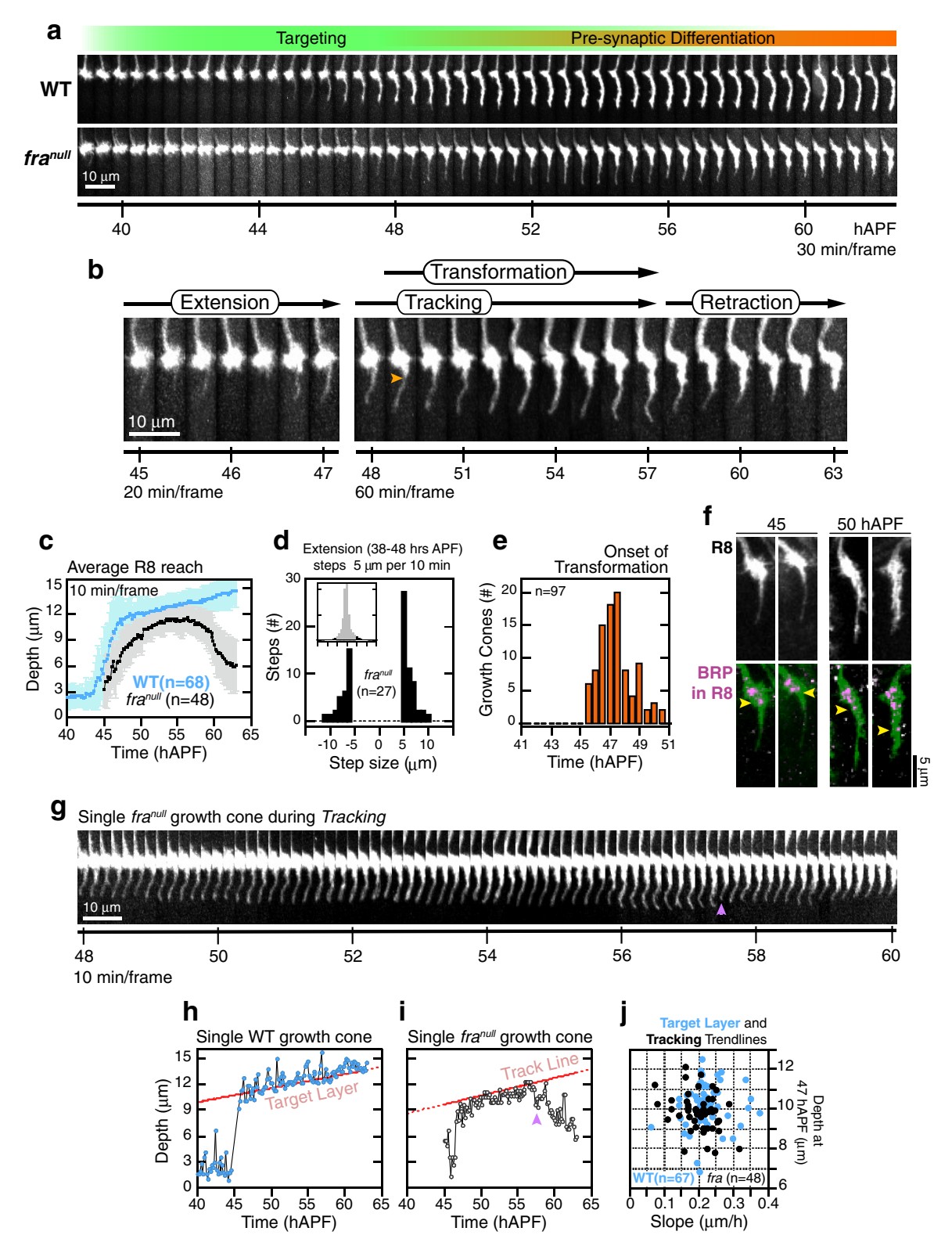

**Figure 5.** *fra^null* R8 targeting. (**a**) Wild-type and *fra^null* growth cones from the same mosaic brain. (**b**) Steps of *fra^null* targeting. Orange arrowhead marks the onset of *transformation*. (**c**) Data for WT and *fra^null* R8s from the same brain presented as in *Figure 2b*. Note that the apparent difference between WT and *fra^null* reach prior to retraction is due to the opposing effects on this population average metric of (1) transient extensions beyond the target layer in the WT (*Figure 3—figure supplement 1b*) and (2) the retractions of mutant growth cones during *tracking* (see text). For other datasets in which

*Figure 5 continued on next page*

*Figure 5 continued*

the difference in average reach is less pronounced, see *Figure 5—figure supplement 1c,d*. (d) Data for *fra^{null}* R8s presented as in *Figure 2c*. (e) Data for *fra^{null}* R8s presented as in *Figure 2d*. (f) Brp accumulation follows anterograde expansion (yellow arrows) during *transformation* in *fra^{null}* growth cones. Panels show confocal images of *fra^{null}* R8 growth cones in MARCM brains at 45 and 50 hAPF. R8s are labeled with myr::GFP and myr::tdTOM (MARCM label, not shown). R8s express V5-tagged Brp using the STaR system (*19*). Overlay of the Brp channel with a mask of the GFP channel highlights R8-localized puncta in magenta. (g) *fra^{null}* dynamics during *tracking;* growth cone in (b) reproduced at full time resolution. Magenta arrowhead marks a transient retraction. See *Figure 5—figure supplement 2* for more examples. (h) Tip trace of the WT growth cone in (a), plotted at 10 min resolution. The target layer, in red, is calculated as described in Materials and methods. (i) Tip trace of the *fra^{null}* growth cone in (b) and (g), plotted at 10 min resolution. Magenta arrowhead marks the transient retraction shown in (g). The track line, in red, is calculated as described in Materials and methods. (j) Scatter plot of slopes and positions of target layer and tracking trendlines at 47 hAPF, for the WT and *fra^{null}* growth cones in (c). The 47 hAPF position is a surrogate for the y-intercept of the trendlines.

The following source data and figure supplements are available for figure 5:

**Source data 1.** Contains numerical data plotted *Figure 5c,d,e,h,i,j*.

**Figure supplement 1.** *fra^{null}* R8 targeting.

**Figure supplement 1—source data 1.** Contains numerical data plotted in *Figure 5—figure supplement 1a,b,c,d,e*.

**Figure supplement 2.** *fra^{null}* dynamics during tracking

---

*Trim9^{null}* single mutant R8s were indistinguishable (*Figure 7e*). Thus, *Trim9* and *Fra* are in the same pathway for R8 targeting.

We analyzed how *Trim9^{null}* targeting defects arise using live imaging. Like fra, *Trim9^{null}* R8s exhibit wild-type *extension* dynamics, their thin targeting processes reach depths comparable to their wild-type counterparts at the *stabilization* step (*Figure 7f*), and the onset of *transformation* occurs on schedule (*Figure 7g*, orange arrowheads). By contrast to *fra^{null}*, we observe tip expansion in ~90% of *Trim9^{null}* R8s (*Figure 7g*, orange barbells, closed ends). The expanded tips of *Trim9^{null}* R8s frequently collapse (82%; *Figure 7g*, orange barbells, open ends)—an event never seen in the wildtype after *stabilization*. Some of these tips exhibit multiple rounds of expansion and collapse. Tip collapse was also commonly observed in mutant R8s that had apparently completed *transformation* (*Figure 7g*, bottom panel). In *fra^{null}* R8s, tip expansion was never observed and this correlated with a lack of stable target layer adhesion. This, together with the observation that the tips of most *Trim9^{null}* R8s expand but nearly all subsequently retract from the target layer (*Figure 7b*), suggest that Trim9 acts downstream of Fra to consolidate the attachment of R8 growth cones to the target layer.

During the time interval corresponding to *elongation* in wildtype, *Trim9^{null}* R8s exhibit one of three types of behavior (*Figure 7f*). Retracting *Trim9^{null}* R8s (30%) are distinguished from *fra* mutants during this period only by their unstable expanded tips (*Figure 7g*, second panel from top); this class *tracks* the target layer up to ~55 hAPF and retracts within the time of observation. Stalled *Trim9^{null}* R8s (30%) are similar to the retracting class except that they maintain the depth they have achieved by ~55 hAPF (*Figure 7g*, third panel from top). The last class, elongating *Trim9^{null}* R8s (40%), appear to complete *transformation* and *elongate* along with their wild type counterparts (*Figure 7g*, bottom panel). As virtually none of the mutant R8s are found at the appropriate target depth in the adult (*Figure 7b*) and tip

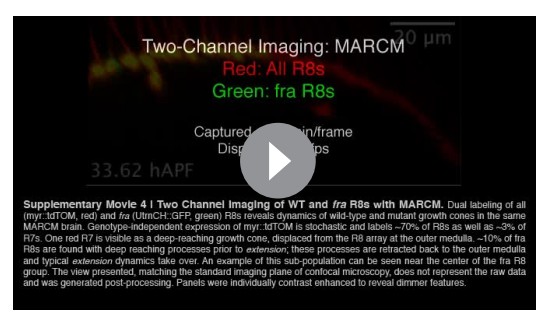

**Video 4.** Two Channel Imaging of WT and fra R8s with MARCM. Dual labeling of all (myr::tdTOM, red) and *fra^{null}*(UtrnCH::GFP, green) R8s reveals dynamics of wild-type and mutant growth cones in the same MARCM brain.

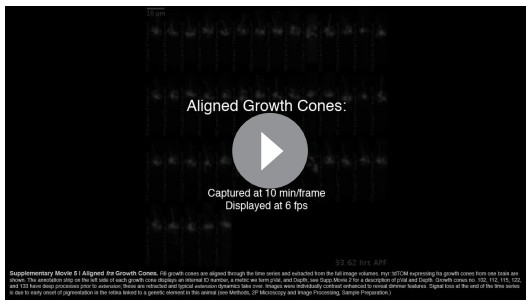

**Video 5.** Aligned *fra^null*Growth Cones R8 growth cones are aligned through the time series and extracted from the full image volumes. myr::tdTOM expressing *fra^null*growth cones from one brain are shown.

collapse was observed in 78% of the terminals during development, this last class of *Trim9^null* R8s must retract from the target layer sometime between 65 hAPF and eclosion (100 hAPF). We note that, similar to what is seen in the *Net* null background (*Figure 6a*), the efficiency of antero-grade progression of *transformation* correlates with the three classes—that is, retracting *Trim9^null* R8s show minimal *transformation* while in all elongating *Trim9^null* R8s the process appears complete. We do not know the factors that influence *transformation* efficiency. While the onset of *transformation* is independent of *Fra*, *Net*, and *Trim9*, it is possible that the anterograde progression of *transformation* is linked to events regulated by these genes, such as tip expansion and target layer adhesion. We suspect, given that *transformation* represents pre-synaptic differentiation, that the extracellular contacts made during this process prevent complete retraction of the mutant R8 processes back to M0, producing the expressivity spectrum observed in all three genes examined.

The analysis of *Trim9^null* supports the conclusion that the Net-Fra pathway regulates adhesion of R8 growth cones to their targets (*Figure 8a*).

## Discussion

Net-DCC is the prototypical ligand-receptor pair required for neural circuit assembly with known roles in diverse processes such as chemoattraction (*Serafini et al., 1996*), target recognition (*Timofeev et al., 2012*), and synaptogenesis (*Colón-Ramos et al., 2007*). Beyond the nervous system, Net signaling has been implicated in cell polarization (*Ziel et al., 2009*) and adhesion (*Srinivasan et al., 2003*; *Yebra et al., 2003*). The details of how Net-DCC affects development will, of course, be context dependent. While this complicates generalization, the diversity of developmental contexts provides different windows into studying this evolutionarily conserved signaling module. R8 is a superb example of this.

We sought to leverage the relative simplicity of R8 geometry and development, and the large numbers offered by the visual system to gain insight into how Net-DCC functions in neural circuit assembly through live imaging. Loss of function mutations in the genes for the ligand, the receptor, and a protein associated with the receptor result in strong end-point phenotypes that are consistent with lack of chemoattraction in a gradient, absence of target recognition, or failed synaptogenesis.

Direct visualization of live mutant and wild-type growth cones in genetically mosaic animals allowed us to distinguish between these possibilities. We argue that in R8 neurons, the Net-Fra pathway promotes attachment, not chemoattraction or target recognition. The onset of synaptogenesis is also Net-Fra independent, and while standard genetic analysis cannot rule out a subsequent role for Net-Fra in this process, the attachment function precedes synaptogenesis. Each R8 growth cone in all three mutant backgrounds studied—*Net*, *fra*, and *Trim9*—extends a single process which reaches the M3 layer as in wild type. The targeting defects observed in the adult arise from retraction from M3 as a consequence of the lack of the stable association and morphological transformation of the tip of the R8 growth cone at the peak of the Net gradient. We note that given the cellular and molecular complexity of the medulla neuropil, retraction may also reflect the presence of repulsive factors in M3 uncovered through loss of Net-based attachment. This, however, is not consistent with the *tracking* behavior of mutant R8s; the persistent association of the tip of R8 growth cones with the target layer (*Figure 5—figure supplement 2*) is difficult to reconcile with a repulsive cue at M3. Despite this and other possible contributions to the terminal phenotype, the simplest interpretation of our data is that the Net-DCC interaction promotes adhesion. For the remainder of the discussion, we use this term to describe the role of Net-DCC signaling in R8 targeting.

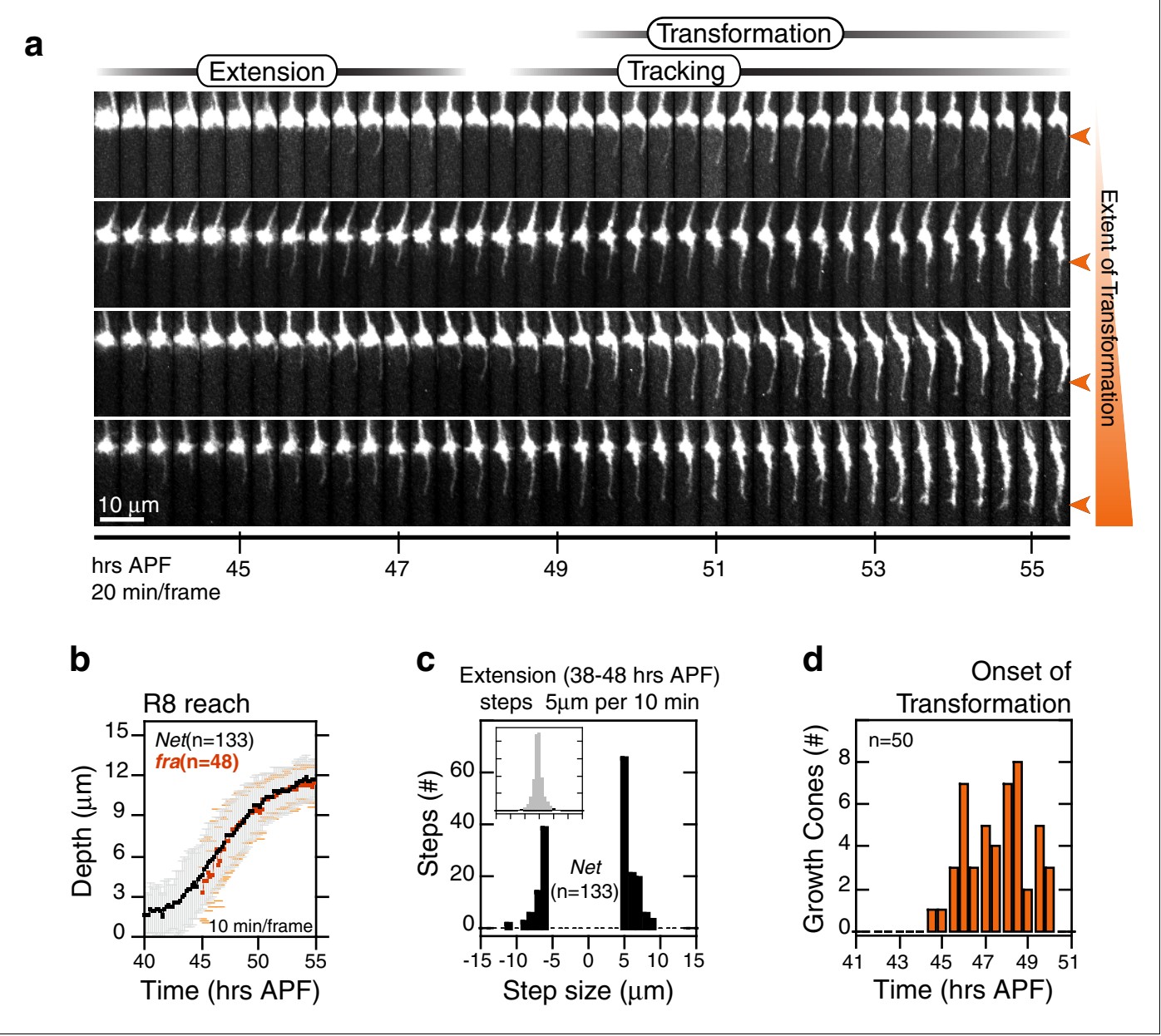

**Figure 6.** R8 targeting in *Net* mutants. (**a**) Four growth cones from the same *Net* null mutant brain. Orange arrowheads mark the extent of *transformation* at the end of the time series in this data set; this determines the final R8 depth after retraction of the thin process. (**b**) Data for R8s in a *Net* mutant brain presented as in *Figure 2b*. The corresponding trace for *fra^{null}* R8s from *Figure 5c* is reproduced in orange. (**c**) Data presented as in *Figure 2c*. (**d**) Data for presented as in *Figure 2d*.

The following source data is available for figure 6:

**Source data 1.** Contains numerical data plotted in *Figure 6b,c,d*.

## Importance of live imaging to explore mechanisms of axon guidance

Live imaging coupled with the unique features of R8 targeting and comparative analysis of wild-type and mutant growth cones in mosaic animals were essential to pinpoint the requirement for the Net-Fra ligand receptor pair for target adhesion. Specifically, the identification and analysis of the *tracking* behavior of mutant R8s would not have been possible in fixed preparations. Standard fixation techniques do not preserve *tracking* filopodia at their full lengths; the shortcomings of fixation have

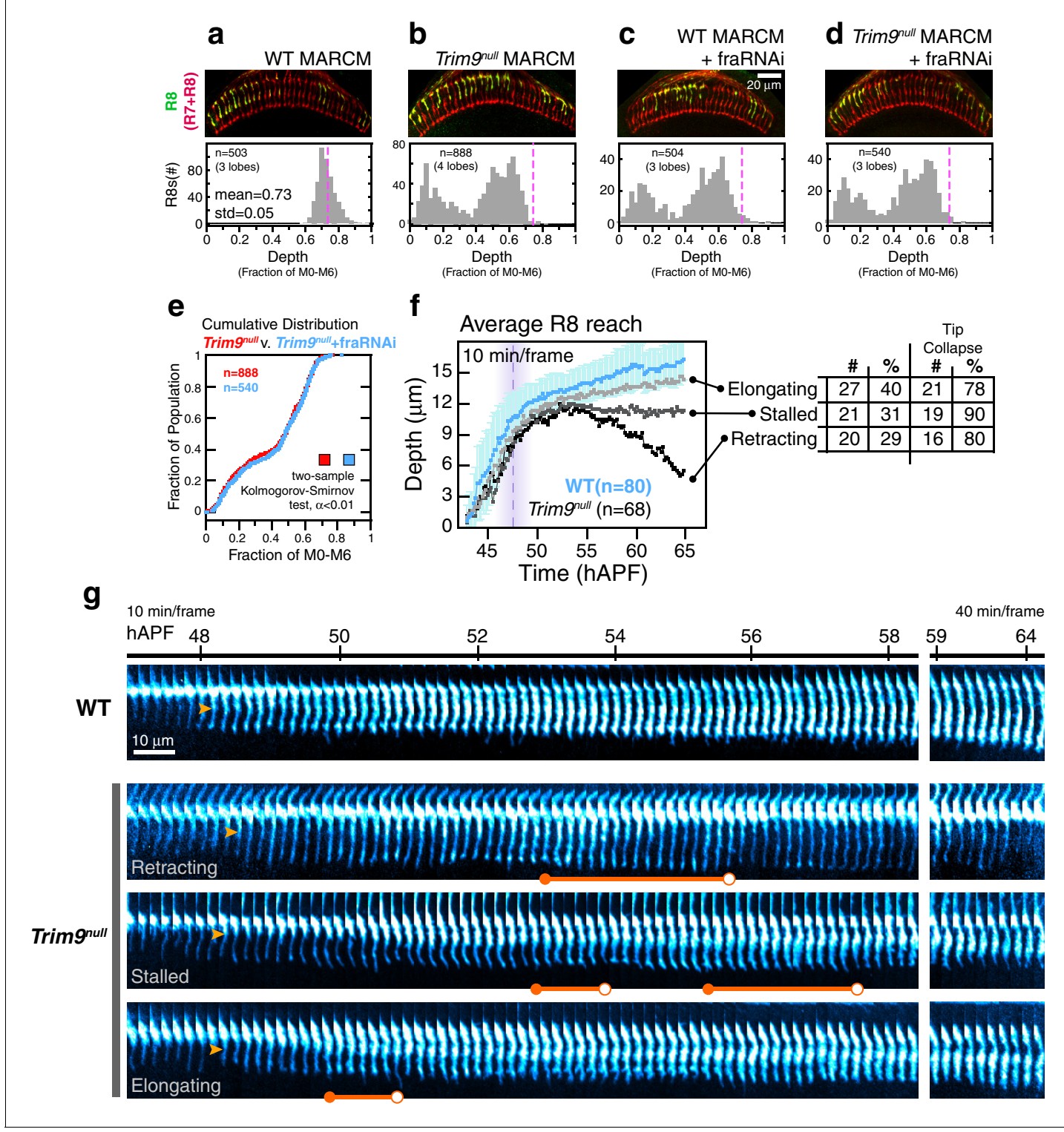

**Figure 7.** *Trim9null* R8 targeting. (**a**) *Panel:* Confocal micrograph of outer medulla in a WT MARCM brain. MARCM labeled WT R8s (green, expressing UtrnCH::GFP) and all R cells (red, labeled with Mab24B10) are shown. *Graph:* Normalized R8s projection depths. Dashed pink line marks the mean of the distribution. (**b**) *Panel:* Confocal micrograph of outer medulla in a *Trim9null* MARCM brain. Cell labeling as in (**a**). *Graph:* Normalized R8s projection depths. Dashed pink line marks the mean of the WT distribution from (**a**). (**c**) *Panel:* Adult brain in which *Fra* expression was knocked down in R8s using a cell-type specific driver (i.e. fraRNAi) in the WT MARCM setup. *Graph:* Normalized R8s projection depths. Dashed pink line marks the mean of the WT distribution from (**a**). (**d**) *Panel:* Adult brain in which *Fra* expression was knocked down in R8s using a cell-type specific driver (i.e. fraRNAi) in the *Trim9null* MARCM setup. *Graph:* Normalized R8s projection depths. Dashed pink line marks the mean of the WT distribution from (**a**). (**e**) Cumulative

*Figure 7 continued on next page*

*Figure 7 continued*

distribution of data in (b) and (d). The distributions are not distinguishable by the two-sample Kolmogorov-Smirnov test at significance level α < 0.01. (f) Reach of the tip into the medulla for WT and *Trim9^null^* R8s, compiled from 2 brains. Error bars are standard deviation for WT. Dashed magenta line and band mark *stabilization*. Table shows number and percentage of mutant R8s in each class and for tip collapse events. As virtually all R8 terminals in the adult fall short of the target layer, the *elongating* class of R8 terminals must retract after 65 hr APF. (g) Wild-type and *Trim9^null^* growth cones from the same mosaic brain. Images are shown with a cyan-hot look-up table to increase displayed dynamic range. Orange arrowheads mark the onset of *transformation*. Orange barbells mark prominent expanded tips (closed end) that collapse (open end).

The following source data is available for figure 7:

**Source data 1.** Contains numerical data plotted *Figure 7a,b,c,d,e,f*.

been highlighted previously for cytonemes (*Kornberg and Roy, 2014*) in the context of the response of cells to morphogens. In addition, the continuous, extended observation of individual growth cones was critical to recognizing general trends over transient dynamics. For example, even with perfect preservation, only half of the *tracking* R8 processes in *fra* mutants would appear to be at the target layer at any given time (*Figure 5—figure supplement 1e*). The simplest interpretation of such data would support the erroneous conclusion that half the R8 population required Net-Fra for chemoattraction. Thus, live imaging allowed us to discriminate between chemoattraction, target recognition, and retraction after reaching the target.

## Net-DCC mediated adhesion in neuronal development

The canonical role of Net-DCC in neuronal development is axon guidance through chemoattraction: a gradient of soluble Net steers DCC-expressing growth cones to their targets. Adhesion is also a recognized output of Net signaling, particularly in non-neuronal systems (*Srinivasan et al., 2003*; *Yebra et al., 2003*). Biochemical and cell biological studies using dissociated vertebrate neurons or neural explants suggest that these different functions may share a fundamental mechanistic similarity. Structurally, Net can be described in three domains. The two N-terminal domains interact with cognate receptors, including DCC (*Keino-Masu et al., 1996*). The C-terminal domain of Net is highly charged and mediates non-specific adsorption to cell surfaces, ECM components, and to tissue culture substrates (*Kappler et al., 2000*; *Moore et al., 2012*; *Yebra et al., 2003*). *In vitro*, substrate-bound Net promotes adsorption of DCC-expressing cells (*Shekarabi et al., 2005*) and the Net-DCC interaction can withstand measurable pulling forces (*Moore et al., 2009*). Net's 'stickiness' can be reduced by masking the charge with heparin or removing the C-terminal domain altogether (*Moore et al., 2012*). These manipulations do not alter the ligand's interaction with DCC, but significantly reduce the neuronal response to Net in classic *in vitro* assays of Net-DCC function, including growth cone expansion, axon outgrowth, and turning (*Moore et al., 2012*). For example, in the explant turning assay, a cluster of Net-secreting cells abutting a dissected embryonic spinal cord can attract extending commissural axons over a distance of some 200 μm (*Kennedy et al., 1994*). Without the C-terminal domain, the range over which Net can attract axons is markedly reduced (*Moore et al., 2012*). Consistent with the importance of substrate binding to the function of externally supplied Net, most of the endogenous Net in the developing mouse spinal cord is cell- or ECM-bound (*Kennedy et al., 2006*). Together, these observations support the notion that "growth cones pull directly on the cues that guide them" (*Moore et al., 2012*), thereby unifying the chemoattraction and adhesion roles of Net-DCC in the concept of haptotaxis, or motility through traction.

The adhesion function of Net-DCC *in vivo* is well characterized in the anchor cell (AC), a non-neuronal cell in *C. elegans*. During development, the AC polarizes toward and invades the basement membrane, initiating the attachment of the uterus to the vulva. UNC-6 – UNC-40 signaling is required for the efficient completion of this process. In a series of studies using live imaging, Sherwood and colleagues demonstrated that UNC-6 localized to the basement membrane stabilizes UNC-40 clustering in the AC, which, in turn, directs polarized F-actin production to the basal surface of this cell (*Hagedorn et al., 2013*; *Wang et al., 2014*; *Ziel et al., 2009*). Without UNC-6, UNC-40 can still form clusters and promote actin assembly, but these patches of activity are transient and randomly positioned around the cell periphery (*Wang et al., 2014*). MADD-2, the *C. elegans* homolog of Trim9, is also required in the AC to maintain stable and appropriately polarized F-actin

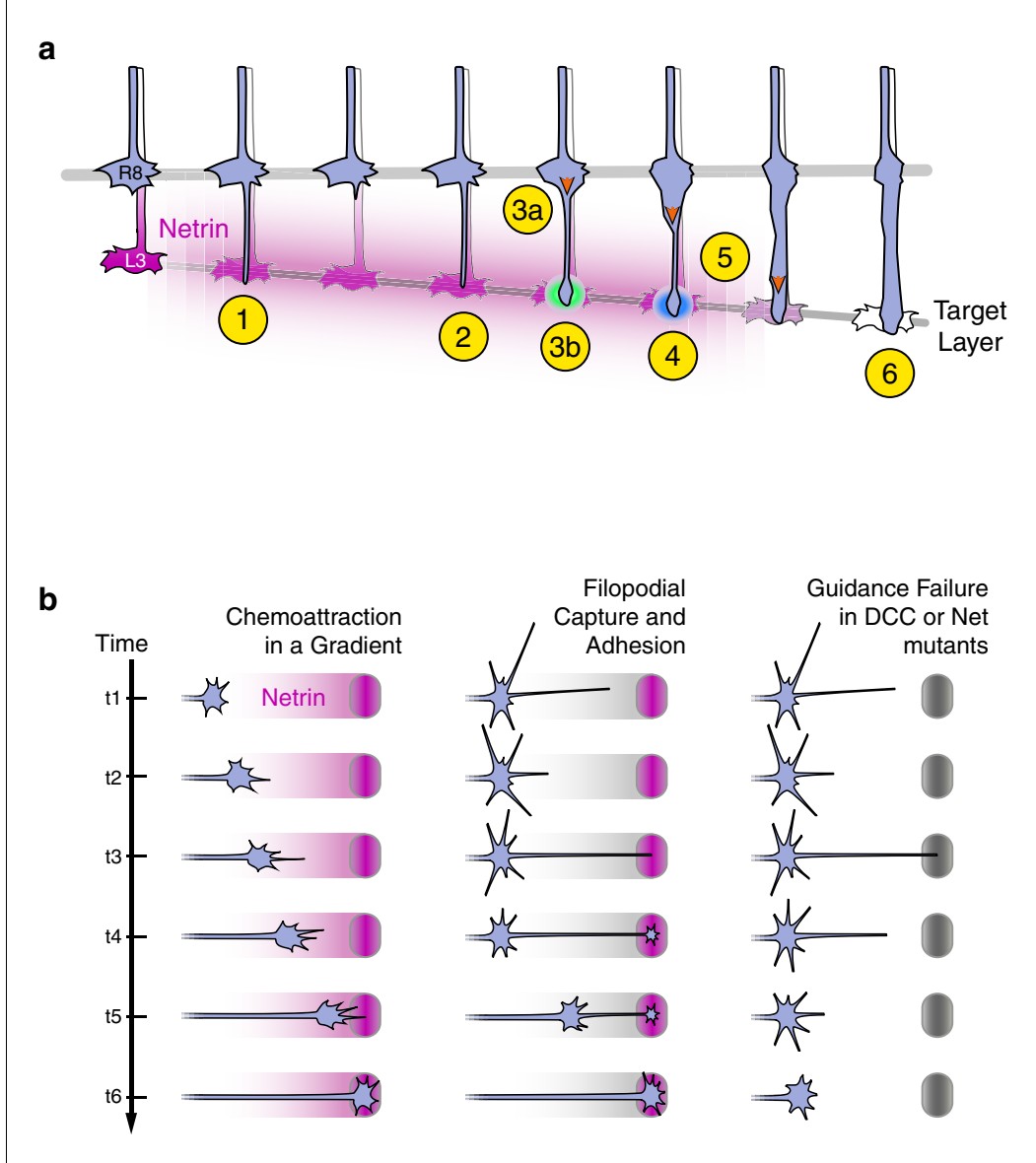

**Figure 8.** Axon guidance through Net-DCC-mediated adhesion. (**a**) Net-Fra signaling in the second step of R8 targeting: (1) *Extension* of thin process into medulla, Net-Fra independent. (2) Target layer recognition, Net-Fra independent. (3a) Onset of *transformation,* proximal expansion (orange arrowhead), Net-Fra independent. (3b) Tip expansion in the target layer, Net-Fra <u>dependent</u> (cyan highlight). (4) Consolidation of adhesion to the target layer, Net-Fra-Trim9 <u>dependent</u> (blue highlight). (5) Progress of *transformation* (orange arrows). The extent of *transformation* completed before the end of *tracking* in the mutant backgrounds may underlie the difference in expressivity between the *Net, fra^{null}*, and *Trim9^{null}* adult phenotypes. (6) *Elongation* and maturation. (**b**) Net-DCC mediated guidance works through target adhesion, not chemoattraction in a gradient. First Column: In the classical view of chemoattraction, DCC-laden filopodia sense the gradient of extracellular Net (pink) and direct the movement of the growth cone toward the source. Second Column: In the alternate model, the Net source is detected independent of the gradient (gray), through filopodial search and capture (t1-t3) or directed extension of a targeting filopodia guided by a separate mechanism (not depicted). Net-DCC signaling at the ligand peak promotes attachment to the target (t4); the growth cone proper then reaches the target with the aid of this initial anchor (t5-t6). Third Column: The generalized view of axon guidance failure in DCC-Net mutants shows that reaching the target does not require the receptor or the ligand; the final phenotype is due to retraction instead of a loss of attraction.

patches; without MADD-2, basal patches can be transient, mis-localized actin assembly is observed, and, ultimately, basement membrane invasion is compromised (*Wang et al., 2014*). The parallels between the use of Net-DCC-Trim9 in development by R8 and the non-neuronal AC—to effect a spatially constrained adhesion or polarization response at an acute presentation of Net (i.e. gradient peak for R8 and basement membrane enrichment for the AC)—raises the possibility that adhesion may be a widely conserved cell biological output of this signaling module.

In this context, R8 targeting and midline crossing in *Drosophila* may function in a fundamentally similar way through an adhesion-based mechanism. Brankatschk and Dickson demonstrated that the growth cones of commissural neurons, like those of R8, respond to a membrane-tethered form of Net in a fashion indistinguishable from the secreted form of Net (2006). If commissural axons can, indeed, reach the midline without this signaling module—that is, if the mutant phenotypes, as in the R8 system, arise due to retraction rather than a lack of attraction, the role of Net-Fra would be better described as enabling the axons to traverse the dense midline neuropil. Alternatively, commissural growth cones may extend towards the midline via haptotaxis, growth via traction (and hence adhesion) along a pathway of increasing levels of Net bound to the surface of midline glia or associated ECM. Thus, given our current state of knowledge of midline crossing, it is possible that Net-Fra act through an adhesive mechanism in this system.

Evidence consistent with this view comes from the work of Emoto and colleagues, who carried out a detailed analysis of Trim9 function in the C4da neurons of *Drosophila* (*Morikawa et al., 2011*). The branched axon terminals of these sensory neurons send projections that cross the midline of the ventral nerve cord (VNC), the larval counterpart of the embryonic midline. In *Net*, *fra*, and *Trim9* mutants, these projections are severely reduced or missing. There are three different sub-classes of C4da neurons and each subclass elaborates a characteristic number of contralateral projections. Neurons with more contralateral projections express higher levels of Trim9. Furthermore, overexpression of Trim9 produces ectopic projections in a dose-dependent manner. This overexpression phenotype is completely suppressed in a *fra*$^{null}$ background (*Morikawa et al., 2011*). Given what we know of Trim9 function in stabilizing adhesion downstream of Net-Fra in R8s and polarization in the AC, the C4da results are consistent with Trim9 controlling the fine-tuning of midline crossing probability. The Net-Fra initiated midline contacts would then be either stabilized or lost depending on Trim9 levels, resulting in successful or retracted contralateral projections, respectively.

These observations in *Drosophila* as well as *in vitro* experiments with vertebrate growth cones raise the interesting possibility that Net-DCC-mediated adhesion may be a common mechanism used by growth cones in developing invertebrate and vertebrate brains.

## Circuit assembly through contact dependent processes

*Tessier-Lavigne and Goodman (1996)* divided guidance signals into four categories. Short-range signals would act through direct contact between growth cones and cells or growth cones and the ECM to attract or repel growth cones. Conversely, long-range signals, secreted by cells, were proposed to act at a distance in a graded fashion to attract or repel growth cones.

In *Drosophila*, both at the midline and in the visual system, a membrane-tethered form of Net is sufficient to rescue guidance phenotypes (*Brankatschk and Dickson, 2006*; *Timofeev et al., 2012*). These results have been interpreted as evidence for a local or short-range function for Net-DCC. This is by contrast to the diffusible gradient view supported by the original conception of long-range chemoattraction and the results of turning assays of vertebrate growth cones *in vitro*. However, both at the midline and in the medulla, tethered Net is present as a gradient that peaks at the targets of guidance, and in the absence of studies testing the relevance of this cell-bound gradient to guidance, the distinction between the short- and long- range functions of Net-DCC remained unclear. Net is a secreted molecule of the laminin superfamily and, in principle, can diffuse to form a gradient. As noted above, however, data from *in vitro* studies argue in favor of the relevance of immobilized Net over that of a soluble form. Together, these results raise the possibility that Net-DCC guidance relies principally on immobilized Net, thereby eroding the mechanistic distinction between the short- and long-range guidance in a gradient.

In the R8 system, the gradient is not required. Our results show that the R8 growth cone can reach and recognize its target without the ligand gradient, or the receptor, in a fashion indistinguishable from wild type, and that Net-DCC act to promote adhesion at the source of Netrin. Despite the presence of Net along the path to M3, we cannot detect an effect of the ligand-receptor interaction

while the tip of the R8 growth cone is moving up the Net gradient (*Figure 5—figure supplement 1b*). Such spatial specificity in signaling output may be achieved through fine-tuning the response of the system to the ligand concentration, or by inhibiting the activity of the receptor until the target is reached.

How R8 uses Net-DCC may be idiosyncratic to this neuron. That is, the evidence against gradient-based chemoattraction in R8 targeting does not rule out a role for this mechanism in other *in vivo* contexts. If Net does act in a graded fashion in other systems, we suggest, in agreement with the work from the Sheetz lab, that it does so not as a soluble factor, but rather as a substrate-bound molecule that influences growth cone behavior through contact. The distribution of the ligand may be determined by localized expression at a single source followed by diffusion and adsorption. Alternatively, a track of cells may form a gradient by expressing and presenting different levels of Net on their surfaces. Thus at a mechanistic level, we propose that, *in vivo*, DCC responds to immobilized, not soluble, Net. This initial adhesive interaction may stabilize attachment as in the case of R8 or, alternatively, this may promote traction for growth cone motility along a surface (i.e. haptotaxis).

This and other studies of Net-DCC signaling in invertebrates and vertebrates raise the possibility that Net and other secreted signals within the developing CNS may act principally in close proximity to the source cells, either associated with the surface of cells or the ECM, to elicit discrete and localized responses. In the absence of action at a distance, neural circuit assembly would proceed in a stepwise fashion where neuronal processes sample a complex environment through direct contact (i.e. via 'touch' rather than 'smell'), integrate these signals and transform them into morphological and biochemical specification.

## Materials and methods

### Histology and confocal microscopy

Histology was performed as described previously (*Chen et al., 2014*) with minor modifications. After antibody incubations, brains were washed into PBS with 0.5% Triton X-100 (PBT). To minimize tissue shrinkage, the brains were moved from PBT to mounting medium (EverBrite, Biotium) through a series of mixtures with increasing concentrations of the latter. The following primary antibodies were used: chicken Pab α-GFP (abcam, ab13970, RRID:AB_300798, 1:1,000), Mab24B10 (*Van Vactor et al., 1988*) (DSHB, RRID:AB_528161, 1:20), rabbit Pab α-DsRed (Clontech, Cat# 632496, RRID:AB_10013483, 1:200), mouse α-V5 (Serotec, Cat# MCA1360, RRID:AB_322378 1:200). The following secondary antibodies were used: Alexa Fluor 488 goat α-chicken, Alexa Fluor 568 goat α-rabbit, Alexa Fluor 647 goat α-mouse (ThermoFisher, Cat# A-11039, RRID:AB_2534096, Cat# A11036, RRID:AB_10563566, Cat# A-21235, RRID:AB_2535804, 1:500). Confocal images were acquired with a Zeiss LSM780 system.

### Confocal image analysis

To carry out 3D measurements in the medulla, imaged volumes are deconstructed into oriented medulla columns bounded by computed surfaces for M0 and M6. A typical multi-channel confocal stack of the medulla measures 150 x 150 x 180 μm and has voxel dimensions of 0.29 x 0.29 x 0.4 μm. In a pre-processing step, all channels are scaled in the z-dimension to achieve unit voxel aspect ratio. The Mab24B10 channel (i.e. all R cells) is used in the deconstruction. R7 axon terminals are the deepest-reaching visible features; a mask of the R7 axon tips is generated by manually cleaning up the image. Local intensity maxima in the original image are selected with this mask and the resulting point cloud is used to define the continuous 3D surface of the M6 layer. The bundles of R cell axons that stretch across the outer medulla surface increases the complexity of the image at M0; a more involved approach is required to define this layer. The area bounding the footprint of the R7 projections in M6 is divided into 50–72 regions, and, for each region, columnar volumes orthogonal to the M6 surface are extracted from the image. Intensity profiles along the column axis of these volumes are used to find local peaks, which are classified according to magnitude and their distance from the M6 surface. These two parameters are used to identify the peaks that reside in the M0 layer; the M0 surface is built from a point cloud derived from the selected peaks. A similar approach is used to compute the Dm4 surface. To define the medulla columns, a sequence of masks laminating the space between the M0 and M6 surfaces are used to generate maximum intensity projections (MIPs)

of the Mab24B10 image. The cross-sections of R7-R8 projections are identified as local intensity maxima in these MIPs. Across the MIP sequence, the intensity maxima are grouped into individual tracks that define the position and orientation of the medulla columns. This information is used to create single panel MIPs of individual R8 axons from the R8 channel. The tips of R8 projections are marked manually on the MIPs and the input is used to calculate 3D distances to the M0 and M6 surfaces. Up to 500 R8s per medulla were scored with this approach. This analysis was written in Matlab (Mathworks) with a critical script sourced form the Mathworks File Exchange repository (*D'Errico, 2005*). Fiji (ImageJ) (*Schindelin et al., 2012*) was used for user-assisted tasks.

## Gaussian mixture modeling of adult R8 phenotypes

Each data set in *Figure 4b–e* was modeled as mixtures of 2–7 Gaussian distributions using a built-in Matlab (Mathworks) function. For all cases, the Akaike information criterion, a fitting evaluation metric that weighs the goodness-of-fit against the number of free parameters, decreased monotonically until the minimum was reached at 4 or 5 components. While more complex models are supported by the data, we presented the fits with 3 components, the minimum number required to represent the major sub-populations. When modeled with >3 Gaussians, only the fra*RNAi* (*Figure 4c*) data supports a wild-type component with a mean at 0.75.

## 2P microscopy and image processing

### Overview

Using a custom-built two photon microscope, we can image the developing visual system between 15 and 65 hr after puparium formation (hAPF). In the specific case of the R8 photoreceptor cell, one imaging session can capture the dynamics of 50–200 individual growth cones at 5–30 min time resolution. We analyzed the 3D time series using a suite of custom scripts implemented in Matlab (Mathworks). The source code for this suite of scripts are available as a supplement to this article.

### Microscope

The microscope was designed to maximize light collection efficiency. The three principal considerations were: (1) High efficiency GaAsP detectors (Hamamatsu); (2) A short, wide-angle collection path with 2" optical elements; and 3. Use of a large field-of-view objective (Zeiss, W Plan-Apochromat 20x/1.0 DIC) that balances a long working distance with a large numerical aperture. A tunable Ti: sapphire pulsed laser (Chameleon Ultra II, Coherent) was used as the light source. Most of the images presented in this study were collected at ~25 mW (920 or 970 nm) under-the-objective power to minimize photobleaching. The microscope hardware and image capture was computer controlled and driven by ScanImage (*Pologruto et al., 2003*).

### Sample preparation

During pupal development, a small, fat-free window over each retina provides optical access into the visual system. The cuticle around the head is removed after head eversion (~12 hAPF), and the animal is attached eye-down on to a coverglass (22 x 50 mm, No.1.5) coated with dilute embryo glue. Up to 18 animals can fit on a sample slide making it possible to image multiple flies in a single session. To provide sufficient immersion liquid for the long imaging sessions, a water reservoir is constructed on the opposite surface of the cover glass. Glass-bound pupae are suspended above a second water reservoir to minimize dehydration. Animals are staged at white pre-pupa formation (0 hAPF) or head eversion (12 hAPF) and kept at 25°C using an objective heater system (Bioptechs).

In all live imaging experiments, genotype-independent R8 labeling was achieved by activating a transcriptionally silenced strong driver with a cell-type specific recombinase (Flp or R, driven by a promoter from the senseless gene, see Experimental Genotypes). The two drivers used, GMR and brp-2A-LexA (see Experimental Genotypes), offered different advantages and caveats. GMR-driven expression of myr::tdTOM provides strong labeling of R8s up to ~50 hAPF. The signal begins to degrade beyond this point due to early onset of pigmentation in the retina and the imaging window closes ~58 hAPF. The modified brp BAC, utilized as a LexA driver, complements the GMR promoter. Expression from brp-2A-LexA dips at ~45 hAPF, but recovers to reveal the details of R8 dynamics past 60 hAPF without significant loss of signal strength. While either strategy yields an adequate

description of WT R8 targeting, combining the strengths of both was essential to a quantitative study of the *fra^null^* phenotype.

## Image processing and analysis

A typical 10 min per frame 24 hr time series contains ~140 512 x 512 x 390 pixel stacks with voxel dimensions of 0.24 x 0.24 x 0.4 μm. In a pre-processing step, these stacks are scaled in the z-dimension to achieve unit voxel aspect ratio. *Medulla registration* begins with manual clean-up of a single time point, the anchor stack (40 hAPF), to remove contributions from the lamina and incoming R8 axons. The cleaned stack is used as a mask to select local intensity maxima in R8 growth cones. The resulting point cloud, which is a sparse representation of the outer medulla surface, is fit to an oblate ellipsoid, yielding the rotation matrix that brings the medulla into alignment with the image axes (i.e. top-down view.) The point cloud itself is aligned to the image axes and used to define the continuous 3D surface of the outer medulla. A mask that contains the R8 growth cones, the shell mask, is built as a slab centered around this surface. Through an iterative cross-correlation search in the rest of the time series, coordinates that most closely match a region-of-interest (ROI) near the center of the R8 array in the anchor stack are identified. An ellipsoidal mask is used to extract volumes-of-interest (VOIs) centered on these coordinates in each stack. Starting from the anchor stack and moving to either end of the time series, the VOIs are iteratively aligned to one another using rigid body transformations. The product of these transformations and the original rotation matrix bring the full time series into register with the anchor stack and align the medulla with the image axes.

*Growth cone segmentation* is performed in a single stack from the registered series, the seed stack (45 hAPF). The seed stack is processed to reduce noise and local intensity variations, masked with the shell mask, and flattened as a MIP. A 2D growth cone template is generated and manually edited to refine the segmentation. The resulting segmented growth cone mask is used to define the 3D center of each growth cone in the seed stack. A refined surface for the outer medulla is computed using the growth cone centers and medulla column vectors for the growth cones are calculated as normals to the new surface.

*Growth cone tracking* is carried out in a MIP representation of the time series, generated from medulla-registered stacks masked with the shell mask. The top-down view of the registered orientation minimizes overlap between R8 growth cones while the shell mask removes signal contribution from non-growth cone objects in the full image volume. Small 2D ROIs centered around each segmented growth cone in the seed stack are used to initiate a cross-correlation-based iterative search to find best matching regions in successive frames of the time series. XY tracks from this search are combined with Z coordinate information retained from the MIP generation step to compile the first-pass XYZT coordinates of the growth cone centers in the registered orientation.

In *growth cone alignment*, the 4D positions of R8 growth cones are refined in the original orientation of the raw data. Starting with the seed stack, a template VOI for each growth cone is extracted from the 3D position and masked with a cylinder oriented along the medulla column vector calculated in the segmentation step. Masked target VOIs are extracted from the next stack in the series, using 3D coordinates from the tracking step and vectors corrected with the medulla transformation matrix. Target VOIs are aligned to template VOIs with rigid body transformations and the aligned target VOIs are re-cast as the templates for the next iteration of the operation (i.e. next stack). This process has two principal outputs: (1) Refined 4D positions and orientations of R8 growth cones, and (2) Masked MIP series of individually aligned growth cones (see *Figure 1d*). To avoid reducing the spatial resolution of the MIP series, rotations about either axis of the imaging plane (XY) are suppressed during MIP generation. As a result, images shown under-represent the true 3D length of the R8 projections (see legend for *Video 2*).

R8 tip position relative to the growth cone center is tracked automatically through each aligned series; this output is visually inspected and corrected, when necessary. The accuracy of automatic tracking increases with image quality and peaks at ~95%. Onset of transformation times were scored manually.

Most image processing was done using custom software written in Matlab (Mathworks). Several critical scripts were sourced from the Mathworks File Exchange repository (*D'Errico, 2005*, *2009*, *Kroon, 2008*). Fiji (ImageJ) was used for batch stack processing and user-assisted tasks.

## Identifying target layer and tracking trendlines

For each growth cone tip trace, a family of candidate trendlines was generated using data points within a moving window of 10–12 hr. For WT growth cones, a subset of the data in each window was selected by fitting a lower-bound cubic spline to the data. The complexity of the spline (i.e. number of cubic functions used) was increased until 30% of the data points were within 0.5 μm of the curve. These spline-proximal points were used to define a line using the Thiel-Sen estimator method. The intercept of the line was adjusted so as to give 90% of all the data in the time window positive residuals. For *fra^null* growth cones, an upper-bound spline was fit to the data in each time window and, again, the complexity of the spline was increased until 30% of the data points were within 0.5 μm of the curve. The best-fit line to the spline-proximal points was refined using a 30% subset with the smallest standard deviation in their residuals to the line. For both classes of growth cones, the optimum trendline from among the candidates was selected using the product of three weight functions. The first two of these are normal distributions, which rank the slope and intercept of the trendlines based on parameters derived from the average tip trace curves of WT growth cones (e.g. blue mean and errors in *Figure 5c*). The third is an exponential decay function that ranks the standard deviation of the residuals to each candidate trendline. This analysis was written in Matlab (Mathworks) with a critical script sourced form the Mathworks File Exchange repository (*D'Errico, 2009*).

## Experimental genotypes

Flies were reared at 25°C on standard cornmeal/molasses medium. Pupal development was staged relative to white pre-pupa formation (0 hAPF) or head eversion (12 hAPF).

### Main figures

**1b,** NetAΔ,NetB::TM/+;; *senseless-R::pest, GMR-RpdOUT-myr::tdTOM/+*

Description: One of the maternal X chromosomes carries a myc-tagged membrane-tethered variant of NetB (NetB::TM) expressed form the native genomic locus of NetB, in a NetA deletion background (*Brankatschk and Dickson, 2006*). Expression of GMR (*Hay et al., 1994*) driven myr::tdTOM is controlled by *sens-R::pest*, resulting in genotype-independent labeling of ~70% of R8s and ~3% of R7s.

**1e,** *W; Sp-Cyo/+; sens-R::pest, brp-RpdOUT-V5-2A-LexA, LexAop-myr::GFP/+*

Description: The R recombinase (*Nern et al., 2011*), under the control of the *sens* F2 fragment (*Pepple et al., 2008*), removes the RpdOUT transcriptional and translational interruption cassette from the modified *brp* BAC (STaR system, RRID:BDSC_55760) (*Chen et al., 2014*). The 3' read-through and translation of the sequence downstream of the excised cassette leads to addition of the V5 epitope tag to Brp (not utilized in this experiment) and production of the LexA transcriptional activator as a discrete polypeptide. The expression domain of the *sens* F2 fragment is specific to R8s in the medulla; LexA driven myr::GFP labels ~70% of R8s and ~3% of R7s.

**2a-d,** same as 1e

**2e,** *sens-FLP1/+; GMR-FRT-Stop-FRT-GAL4/+; brp-FlpdOUT-GFP-2A-LexA, UAS- FRT-Stop-FRT-myr::tdTOM/+*

Description: FLP1 recombinase driven by the *sens* promoter fragment (RRID:BDSC_55768) excises the FRT-flanked transcriptional stop cassettes in the GAL4 and UAS elements, leading to myr::tdTOM expression in R8s. The stop cassette in the UAS element is not necessary for cell-specific labeling. The STaR element is included to visualize Brp puncta in the live preparation (not shown).

**2f,** Same as 1e

**3,** *w; LexAop-myr::tdTOM, R25F07-LexAp65/+; senseless-R::pest, GMR-RpdOUT-myr::tdTOM/+*

Description: R25F07-LexAp65 (RRID:BDSC_52703) drives expression of myr::tdTOM in Dm3 cells through pupal development into adulthood. R8-specific expression of myr::tdTOM achieved as in 1b.

**4a,** *w; Sp-Cyo/+; sens-R::pest, brp-RpdOUT-V5-2A-LexA, LexAop-myr::GFP/+*

Description: see 1e.

**4b,** *ey^{3.5}-FLP1/+; FRT42B, ACT-GAL80/FRT42B, fra^3; sens-GAL4, UAS-UtrnCH::GFP/ sens-R::pest, GMR-RpdOUT-myr::tdTOM*

Description: Variant of the MARCM genotype in 5a. The filamentous actin marker UtrnCH::GFP (*Burkel et al., 2007*) (provided by Margot E. Quinlan) is driven by *sens-GAL4* and positively labels

*fra³* (*Kolodziej et al., 1996*) R8s. Expression of GMR (*Hay et al., 1994*) driven myr::tdTOM is controlled by *sens-R::pest*, resulting in genotype-independent labeling of ~70% of R8s and ~3% of R7s.

**4c**, *w/Y; sens-GAL4/+; sens-R::pest, brp-RpdOUT-V5-2A-LexA, LexAop-myr::GFP/UAS-fra RNAi (DRSC HMS01147)*

Description: The three elements on the third chromosome use the STaR system to label ~70% of R8s with myr::GFP (see genotype description for 1e). *sens*-GAL4 drives expression of the short hairpin RNAi construct directed at *fra* (RRID:BDSC_40826).

**4d**, *w, NetAB^{ΔGN}/Y; sens-GAL4/+; sens-R::pest, brp-RpdOUT-V5-2A-LexA, LexAop-myr::GFP/TM2*

Description: The three elements on the third chromosome use the STaR system to label ~70% of R8s with myr::GFP (see genotype description for 1e). *sens*-GAL4 is included as a genetic background control for the *fra-Net* epistasis experiment.

**4e**, *w, NetAB^{ΔGN}/Y; sens-GAL4/+; sens-R::pest, brp-RpdOUT-V5-2A-LexA, LexAop-myr::GFP/ UAS-fra RNAi (DRSC HMS01147)*

Description: See genotype description for 4d.

**4f**, *Net:* Same as 4d.; *Net+fraRNAi:* Same as 4e

4g,H,i,

WT: *w/Y; UAS-myr::tdTOM/Sp; sens-R::pest, brp-RpdOUT-V5-2A-LexA, LexAop-myr::GFP/R23G11-GAL4*

*Net:* w, *NetAB^{ΔGN}/Y; UAS-myr::tdTOM/+; sens-R::pest, brp-RpdOUT-V5-2A-LexA, LexAop-myr::GFP/R23G11-GAL4*

Description: The three elements on the third chromosome use the STaR system to label ~70% of R8s with myr::GFP (see genotype description for 1e). R23G11-GAL4 (RRID:BDSC_49043) drives expression of myr::tdTOM in Dm4 cells in the adult (*Nern et al., 2015*).

4j,

Dm1-WT: Similar to 4g-WT with R22D12-GAL4 (RRID:BDSC_48983) driving Dm1 specific expression (*Nern et al., 2015*).

Dm1-*Net:* Similar to 4g -*Net* with R22D12-GAL4 (RRID:BDSC_48983) driving Dm1 specific expression.

Dm6-WT: Similar to 4g -WT with R38H06-GAL4 (RRID:BDSC_50029) driving Dm6 specific expression (*Nern et al., 2015*).

Dm6-*Net:* Similar to 4g -*Net* with R38H06-GAL4 (RRID:BDSC_50029) driving Dm6 specific expression.

**5a-c,f-j**, *ey^{3.5}-FLP1/+; FRT42B, ACT-GAL80/FRT42B, fra³; sens-GAL4, UAS-myr::tdTOM/ sens-R::pest, brp-RpdOUT-V5-2A-LexA, LexAop-myr::GFP*

Description: Mitotic recombination in the visual system in this MARCM (*Lee and Luo, 1999*) genotype was driven by FLP1 recombinase (*Nern et al., 2011*) under the control of the *ey-3.5* promoter fragment (*Bazigou et al., 2007*). The pairing of the *ey-3.5* promoter with this higher efficiency variant of FLP largely preserves its specificity in the eye disc; in ~15% of the optic lobes we also noted recombination in lamina and medulla cell precursors. In our analysis, we did not detect a subclass of *fra* R8s with alternate developmental progression, suggesting that the possible existence of sporadic *fra* mutants of other cell classes does not change our main conclusions. The ACT-GAL80 element is based on a new actin-derived pan-cell promoter (provided by Barret Pfeiffer, Rubin Lab, Janelia Farm Research Campus/HHMI). *fra* R8s are labeled with myr::tdTOM (not shown). STaR system is used to drive genotype-independent expression of myr::GFP in ~70% of R8s (see genotype description for 1d).

**5d**, same as 4b

**5e**, Data compiled from two MARCM genotypes:

*ey^{3.5}-FLP1/+; FRT42B, ACT-GAL80/FRT42B, fra³; sens-GAL4, UAS-myr::tdTOM/ sens-R::pest, brp-RpdOUT-V5-2A-LexA, LexAop-myr::GFP* , and

*ey^{3.5}-FLP1/+; FRT42B, ACT-GAL80/FRT42B, fra³; sens-GAL4, UAS-UtrnCH::GFP/ sens-R::pest, GMR-RpdOUT-myr::tdTOM*

**6**, w, *NetAB^{ΔGN}/Y; Sp-+/+; sens-R::pest, GMR-RpdOUT-myr::tdTOM/+*

Description: R8-specific expression of myr::tdTOM achieved as in 1b.

**7a**, *ey^{3.5}-FLP1/+; ACT-GAL80, FRT40A/FRT40A; sens-GAL4, UAS-UtrnCH::GFP/ TM2*

**7b**, *ey^{3.5}-FLP1/+; ACT-GAL80, FRT40A/Trim9^{91}, FRT40A; sens-GAL4, UAS-UtrnCH::GFP/ Ly*

**7c**, *ey$^{3.5}$-FLP1/+; ACT-GAL80, FRT40A/FRT40A; sens-GAL4, UAS-UtrnCH::GFP/ UAS-fra RNAi (DRSC HMS01147)*

**7d**, *ey$^{3.5}$-FLP1/+; ACT-GAL80, FRT40A/Trim9$^{91}$,FRT40A; sens-GAL4, UAS-UtrnCH::GFP/ UAS-fra RNAi (DRSC HMS01147)*

**7e**, *Trim9$^{null}$:* Same as 7b.; *Trim9$^{null}$* + fraRNAi: Same as 7d.

**7f**, *ey$^{3.5}$-FLP1/+; ACT-GAL80, FRT40A/Trim9$^{91}$, FRT40A; sens-GAL4, UAS-myr::tdTOM/ sens-R:: pest, brp-RpdOUT-V5-2A-LexA, LexAop-myr::GFP*

Description: FRT40A variant of the MARCM genotype in 5a.

## Figure supplements

**Figure1-F.s.1,** Same as 1e.
**Figure3-F.s.1,** Same as 3.
**Figure5-F.s.1a,d,** Same as 4b.
**Figure5-F.s.1c,** Same as 5a.
**Figure5-F.s.1b,e,** Same as 5e.
**Figure5-F.s.2,** Same as 5a.

## Acknowledgements

We thank Joshua Trachtenberg for his help in designing and building our 2P microscope. We thank Maya Bader for her contributions to the identification and initial characterization of Trim9 as an R8 targeting determinant. We thank Gerry Rubin, Aljoscha Nern, and Barret Pfeiffer for providing flies containing various cell-type specific GAL4s and the new GAL80 constructs. We thank Margot E. Quinlan for the *UAS-UtrnCH::GFP* DNA and flies. We thank Thomas Kidd for the NetAB$^{\Delta GN}$ fly line. We thank Kazuo Emoto for the *Trim9$^{91}$, FRT40A* fly line. We thank Iris Salecker for the NetB::TM fly line. For critical reading of the manuscript, we thank Samantha Butler, Thomas R Clandinin, Claude Desplan, Matthew Y Pecot, Margot E Quinlan, Wael Tadros, Liming Tan, and Joshua Trachtenberg. This work was supported by a postdoctoral research fellowship from the Damon Runyon Cancer Research Foundation to OA. SLZ is an investigator of the Howard Hughes Medical Institute.

## Additional information

### Funding

| Funder | Author |
| --- | --- |
| Howard Hughes Medical Institute | S Lawrence Zipursky |
| Damon Runyon Cancer Research Foundation | Orkun Akin |

The funders had no role in study design, data collection and interpretation, or the decision to submit the work for publication.

### Author contributions

OA, Conception and design, Acquisition of data, Analysis and interpretation of data, Drafting or revising the article; SLZ, Conception and design, Analysis and interpretation of data, Drafting or revising the article

### Author ORCIDs

S Lawrence Zipursky, http://orcid.org/0000-0001-5630-7181

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
