## [Decision Letter]

Thank you for submitting your article "Netrin-DCC dependent axon guidance works through adhesion in the *Drosophila* visual system." for consideration by *eLife*. Your article has been favorably evaluated by Eve Marder as the Senior Editor and three reviewers, one of whom is a member of our Board of Reviewing Editors. The following individual involved in review of your submission has agreed to reveal his identity: Liqun Luo (Reviewer #2).

The reviewers have discussed the reviews with one another and the Reviewing Editor has drafted this decision to help you prepare a revised submission. We hope you will be able to submit the revised version within two months.

Summary:

The work from the Zipursky laboratory has combined quantitative imaging in the intact *Drosophila* visual system with genetic manipulation of key axon guidance genes to investigate the rules and logic governing the axon guidance and target selection process. All three reviewers are enthusiastic about the work, appreciating the quantitative nature of the data and the novelty of the findings. As such, all three reviewers are enthusiastic about future publication in *eLife*. However, all three reviewers recommend greater caution with respect to data interpretation. The reviewers are not entirely convinced by the argument that these data over-turn existing models of chemo-attraction, in favor of target-dependent adhesion, particularly as adhesion is not directly assessed in the current paper. All three reviewers recommend revision of the text to reflect a more careful approach to data interpretation. In this context, the authors could consider the importance and caveats of existing data in greater depth (see specific comments of reviewers #2 and #3). There is also room for alternative interpretations that are not currently considered (see specific comments of reviewer #1). Clearly, the authors need not cite the examples or ideas proposed by the reviewers, but they should take full advantage of the fact that *eLife* does not impose strict limits on the length of the text. The title must be adjusted to be reflective of the novel ideas contained in the manuscript, without the definitive statement that Netrin acts through adhesion. Then, in generating a more thorough and even argument, the authors are free to favor their existing argument, but it should be done in the context of broader discussion.

Reviews:

The reviews from each of the individual reviewers are included here since they are in complete agreement and present a unified view of the work and make a unified set of recommendations.

*Reviewer #1:*

The work from the Zipursky laboratory is a beautiful example of applying detailed imaging analysis to the questions of axon guidance and target recognition in an intact organism. The authors examine the effects of loss of a well-studied axon guidance molecule and, by examining the time-resolved response underlying failed guidance, arrive at a striking new observation. Essentially, the authors observed that the growth cones reach their target before retracting.

The significance of the claims made in the manuscript is based on the observation that a growth cone retracts only when it reaches the target cell, in mutations that delete Netrin signaling. The authors argue that this refutes gradient-based chemoattractive signaling by Netrin since the phenotype is only observed upon target contact. The counter model, provided by the authors, is that Netrin mediates target-dependent adhesion. In the absence of target-dependent adhesion, the growth cone collapses. This is the basis for a proposed over-turning the chemotactic gradient model. However, there is no direct evidence for this adhesive model in this system, only reference to the potential adhesive properties of Netrin in other systems. The authors adamantly support their model without excluding or entertaining other equally plausible ideas. This is where the authors should be more circumspect.

Growth cone retraction necessitates a counteracting force. A lack of a single adhesive interaction would not necessarily cause growth cone retraction. Indeed, it might be expected to cause a bypass of the target or cause increased growth cone elaboration, as it searched for the correct target. In essence, the growth cone collapse that the authors observe could equally be caused by unveiling a prominent contact-dependent repulsive cue that is normally overcome by the presence Netrin. If so, the observation of target-dependent growth cone collapse in the absence of Netrin is not evidence for a lack of a gradient of Netrin. Thus, in my view, this paper is an important addition to the literature, but does not effectively refute the existing model of Netrin function. Indeed, it is difficult to completely argue against early work in vitro showing secretion of Netrin acting at a distance in cell explants. Thus, it would be appropriate to consider models, such as the one suggested above – or others – that would also be consistent with the fundamental observation In essence, the absence of causal evidence is not evidence in favor their model.

Beyond this, I have little criticism of the data acquisition or analysis. The experiments are nicely performed, the genetics are solid and the image analysis is quite nice. I am enthusiastic about publication in *eLife*.

*Reviewer #2:*

This study made valuable contributions to the axon guidance literature by carefully observing and quantifying the behavior of wild-type and DCC mutant R8 growth cones before and at their medulla targeting M3 zone. The authors found that DCC mutant R8 growth cones reach the M3 zone, but fail to stabilize and eventually retract. Based on these observations, the authors conclude that Netrin/DCC do not act as long-range chemoattraction, but short-range adhesion to stabilize the contact. Overall the technical quality of the data is high and I support its publication in *eLife*.

In their interesting discussion of the Netrin literature, the authors set up two alternative models for Netrin's action: 1) acting locally to promote adhesion (as in the case of R8 axon targeting to the M3 zone); 2) acting at a long-range attractive molecule (as was proposed in the classic case of midline guidance). While I agree with the authors that the evidence for netrin acting as a diffusible long-range signal in vivo remains to be proven, due to technical limitations, its long-range action in in vitro explants is quite strong. For example, in Figure 5 of the Kennedy et al. paper (Cell, 78:425, 1994), floor plate tissues or Cos cells transfected with Netrin, when placed on the side of the dorsal spinal cord explants, can cause commissural axons that otherwise grow ventrally to turn towards the side. Axons as far as 250 micrometers away from the source of netrin-expressing cells can be induced to turn. It is possible that secreted netrin is deposited on the substrate in the dorsal spinal cord explant and not freely diffusible, and axon turning is caused by preferring ever stronger force of adhesion to higher concentration of netrin. In this regard, adhesion and chemoattraction is not mutually exclusive. Adhesion may be a specific mechanism by which axons are attracted to its target. The authors may consider incorporating this in their Discussion.

*Reviewer #3:*

This careful study by Akin and Zipursky examines how the Net/*fra* system works in *Drosophila* R8 growth cone morphogenesis. Although originally proposed to be a long range chemoattractant, numerous works, first in vertebrates and then in *Drosophila*, have indicated that Netrin/DCC can function in cell adhesion. Akin and Zipursky use live cell imaging to examine whether in R8s, net acts as a chemotaxic agent via *fra*. Using several germline and mosaic genetic approaches, the authors demonstrate that the initial extension of the R8 filopodial process towards the net-producing cell is not affected in these mutants. Instead, it is the stabilization of that extension that requires net and *fra*. While adhesion itself is not addressed by any of the experiments, the observations are consistent with a role for Net/*fra* in, not chemotaxis, but stabilization of the R8 projection.

The re-examination of the phenotype, and reevaluation of the penetrance of the Net mutation for R8 targeting is of interest, but the crux of the paper is clearly Figure 5, which describes the correct targeting of the initial R8 process in the absence of *fra*.

This work builds on work on Netrin as well as in other fields, such as the recent work by JP Vincent, to argue that many cues thought to function at a distance may in fact work in direct cell-cell communication. This work also builds on work such as that of Heiman and Shaham indicating that anchoring of a neuronal projection can be an important aspect of neuronal morphogenesis and function.

A few points of clarification: Dm3 is used to define the edge of M3 in Figure 3. Was there a reason to switch from using Dm3 as a fiduciary mark to using all R neurons in Figure 4?

Also, the authors state that the third, deepest component of the R8 distribution for *net* and *fra* mutant cells are "nearly identical" (subsection “Revisiting the *Net-fra* phenotypes”, third paragraph). However, the contribution of the deepest reaching component in Figure 4 seems much more substantial than that in Figure 4, arguing that they are not nearly identical. Figure 4 appear more similar (although not in the first, most superficial component).

A distinction is made between *fra* and Trim9, in that the former is absolutely required for tip expansion, but the latter is not. On this point, Net is glossed over, but Figure 6 appears to show at least one example of tip expansion in the absence of Net. If true, this may also argue for Net-independent functions of *fra*.

---

## [Author Response]

*[…] Summary:*

*The work from the Zipursky laboratory has combined quantitative imaging in the intact Drosophila visual system with genetic manipulation of key axon guidance genes to investigate the rules and logic governing the axon guidance and target selection process. All three reviewers are enthusiastic about the work, appreciating the quantitative nature of the data and the novelty of the findings. As such, all three reviewers are enthusiastic about future publication in eLife. However, all three reviewers recommend greater caution with respect to data interpretation. The reviewers are not entirely convinced by the argument that these data over-turn existing models of chemo-attraction, in favor of target-dependent adhesion, particularly as adhesion is not directly assessed in the current paper. All three reviewers recommend revision of the text to reflect a more careful approach to data interpretation. In this context, the authors could consider the importance and caveats of existing data in greater depth (see specific comments of reviewers #2 and #3). There is also room for alternative interpretations that are not currently considered (see specific comments of reviewer #1). Clearly, the authors need not cite the examples or ideas proposed by the reviewers, but they should take full advantage of the fact that eLife does not impose strict limits on the length of the text. The title must be adjusted to be reflective of the novel ideas contained in the manuscript, without the definitive statement that Netrin acts through adhesion. Then, in generating a more thorough and even argument, the authors are free to favor their existing argument, but it should be done in the context of broader discussion.*

In this final version of the manuscript, we responded to the reviewers’ constructive comments by changing the focus of the Introduction and Discussion from what was perceived as challenge to existing views on chemoattraction by Net-DCC to an attempt to bring together seemingly disparate functions of this fundamental signaling module. While we disagree with the reviewers’ views on our conclusions regarding the adhesion function of Net-DCC in the R8 system, we did remove the word ‘adhesion’ from the title, as suggested. Please find our detailed responses to the specific points raised below. We believe that this review process has improved the manuscript significantly, by challenging us to refine our message and identifying areas that needed clarification.

*Reviews:*

*The reviews from each of the individual reviewers are included here since they are in complete agreement and present a unified view of the work and make a unified set of recommendations.*

*Reviewer #1:*

*[…] The significance of the claims made in the manuscript is based on the observation that a growth cone retracts only when it reaches the target cell, in mutations that delete Netrin signaling. The authors argue that this refutes gradient-based chemoattractive signaling by Netrin since the phenotype is only observed upon target contact. The counter model, provided by the authors, is that Netrin mediates target-dependent adhesion. In the absence of target-dependent adhesion, the growth cone collapses. This is the basis for a proposed over-turning the chemotactic gradient model.*

We recognize that the manuscript at the initial submission stage was focused on drawing a contrast between the accepted view of chemoattraction by Net-DCC and the evidence in the primary literature. In this revised version, we do not directly compare R8 targeting to other examples of Net-DCC dependent axon guidance. We do, however, note that there is a gradient of Net along the path of the R8 growth cone and that neither this ligand gradient, nor the receptor, DCC, are required for reaching or recognizing the target of guidance, the M3 layer. As such, the gradient based chemoattraction model does not apply to R8 targeting. This is a strong conclusion; however, it is limited to the context of R8 targeting and the revised manuscript reflects this scope.

*However, there is no direct evidence for this adhesive model in this system, only reference to the potential adhesive properties of Netrin in other systems. The authors adamantly support their model without excluding or entertaining other equally plausible ideas. This is where the authors should be more circumspect.*

*Growth cone retraction necessitates a counteracting force. A lack of a single adhesive interaction would not necessarily cause growth cone retraction. Indeed, it might be expected to cause a bypass of the target or cause increased growth cone elaboration, as it searched for the correct target.*

Short of mechanical perturbation – a technically impossible experiment for most in vivo systems – we are puzzled as to what would satisfy the reviewer as “direct evidence” for adhesion. Our argument for an adhesive function for Net-DCC signaling at M3 rests on two observations:

1) During the *elongation* stage, R8 axons grow at a rate proportional to the growth of the medulla. The data for this result is shown in Figure 3. The slow and coordinated growth of R8 projections during *elongation* stand in sharp contrast to the fast and individualized dynamics of the *extension* stage. It is possible that the apparently coordinated growth is indeed due to active extension by each R8 projection. However, this extension happens while the growth cones are transforming into nascent axon terminals with pre-synaptic components assembling at future sites of synaptic contacts. This suggests that the R8 projection is no longer a motile growth cone and supports the notion that *elongation* represents passive lengthening in response to the expansion of the surrounding tissue. From this, we infer that R8 projections become attached to the target layer at the onset of *elongation*, i.e. at the *stabilization* step. In the interest of a streamlined main text, much of this detailed account was provided in Supplementary Discussion; it is now part of the main text (subsection “Live imaging reveals that R8 targeting occurs via discrete steps”, third paragraph).

2) In the absence of Net-DCC signaling, the *stabilization* step is lost and *elongation* is replaced by a behavior we described as *tracking*. Briefly, mutant R8 processes continue to extend and retract while they follow a moving position in the medulla that is consistent with the target layer. R8s growth cones exhibit this *tracking* behavior for an average of 10 hours before they ultimately retreat from the target layer. The retractions seen during and after *tracking* indicate that in contrast to wildtype, mutant R8s are not stably attached to the target layer. In addition to this dynamic phenotype, there is a morphological correlate to the loss of *stabilization*: the tips of mutant R8 processes do not expand. We note that there is an interesting parallel between Net-DCC dependent tip expansion in the R8 system and Net-dependent growth cone expansion observed with vertebrate neurons in vitro.

In summary, we interpret the data described above to conclude that wildtype R8s become attached to the target layer at the *stabilization* step and that DCC-Net signaling mediates this attachment.

*In essence, the growth cone collapse that the authors observe could equally be caused by unveiling a prominent contact-dependent repulsive cue that is normally overcome by the presence Netrin.*

We appreciate the reviewer’s caution that growth cone dynamics, particularly in the native environment of the developing CNS, is a very complex process under the influence of many factors and forces. In the manuscript submitted we did note that retraction could, as the reviewer suggested, be a consequence of uncovering of a repulsive factor in the absence of Net. We have now expanded on this (Discussion, third paragraph) to underscore the complexity of phenotypic analysis in the complex environment in the developing CNS. It does seem unlikely that retraction is a consequence of repulsion, however, as it is difficult to reconcile the persistent association of the R8 tip in mutants with the target layer (Figure 5—figure supplement 2) with a contact-dependent repulsive cue. That is, in the absence of DCC *tracking* growth cones often remain associated with target layer for hours. We note that this observation could only be appreciated through live imaging.

*If so, the observation of target-dependent growth cone collapse in the absence of Netrin is not evidence for a lack of a gradient of Netrin. Thus, in my view, this paper is an important addition to the literature, but does not effectively refute the existing model of Netrin function.*

The argument against the relevance of the Net gradient in R8 targeting does not depend on the role of Net-DCC at the target layer. As the reviewer notes, R8s can reach the target layer without the gradient – that is, the terminal mutant phenotype arises from retraction and not a lack of attraction. This is the evidence that argues against chemoattraction in this system.

*Indeed, it is difficult to completely argue against early work in vitro showing secretion of Netrin acting at a distance in cell explants. Thus, it would be appropriate to consider models, such as the one suggested above – or others – that would also be consistent with the fundamental observation In essence, the absence of causal evidence is not evidence in favor their model.*

*Beyond this, I have little criticism of the data acquisition or analysis. The experiments are nicely performed, the genetics are solid and the image analysis is quite nice. I am enthusiastic about publication in eLife.*

We expanded our discussion of the work of Moore and colleagues (Moore et al., 2012) in the revised manuscript (subsection “Net-DCC mediated adhesion in neuronal development”, first paragraph). This study shows that substrate adsorption of Net is important to the neuronal response to Net in many classic in vitro assays, including growth cone expansion, axon outgrowth, and axon turning in embryonic spinal cord explants. While their results do not argue against the diffusibility of Net, they do suggest that the biological output of Net-DCC depends on immobilized ligand. As such, the classic view of chemoattraction by Net-DCC may be better thought of as haptotaxis, or motility through traction, rather than chemotaxis. From this perspective, the adhesion function we infer for Net-DCC in R8s is wholly consistent with the recognized role of Net-DCC in axon guidance.

*Reviewer #2:*

*[…] In their interesting discussion of the Netrin literature, the authors set up two alternative models for Netrin's action: 1) acting locally to promote adhesion (as in the case of R8 axon targeting to the M3 zone); 2) acting at a long-range attractive molecule (as was proposed in the classic case of midline guidance). While I agree with the authors that the evidence for netrin acting as a diffusible long-range signal in vivo remains to be proven, due to technical limitations, its long-range action in in vitro explants is quite strong. For example, in Figure 5 of the Kennedy et al. paper (Cell, 78:425, 1994), floor plate tissues or Cos cells transfected with Netrin, when placed on the side of the dorsal spinal cord explants, can cause commissural axons that otherwise grow ventrally to turn towards the side. Axons as far as 250 micrometers away from the source of netrin-expressing cells can be induced to turn. It is possible that secreted netrin is deposited on the substrate in the dorsal spinal cord explant and not freely diffusible, and axon turning is caused by preferring ever stronger force of adhesion to higher concentration of netrin. In this regard, adhesion and chemoattraction is not mutually exclusive. Adhesion may be a specific mechanism by which axons are attracted to its target. The authors may consider incorporating this in their Discussion.*

We agree with the reviewer’s point that adhesion and chemoattraction are not mutually exclusive and we have included this point in the revised manuscript (subsection “Net-DCC mediated adhesion in neuronal development”, first paragraph and subsection “Circuit assembly through contact dependent processes”, fourth paragraph). We also included a discussion of the turning assay the reviewer describes (subsection “Net-DCC mediated adhesion in neuronal development”, first paragraph) – please see also our last response to Reviewer #1.

*Reviewer #3:*

*[…] A few points of clarification: Dm3 is used to define the edge of M3 in Figure 3. Was there a reason to switch from using Dm3 as a fiduciary mark to using all R neurons in Figure 4?*

The Dm3 cell elaborates its processes at the M2/3 boundary and is used in this study as fiducial layer marker in the distal medulla. In theory, the analysis presented in Figure 4 could have been carried out with this marker. In practice, it would have been technically challenging to incorporate two additional transgenes into all of the different genetic backgrounds studied in Figure 4. The monoclonal antibody 24B10 specifically labels R cells in all genetic backgrounds, which eliminates the need to incorporate further transgenic elements.

*Also, the authors state that the third, deepest component of the R8 distribution for net and fra mutant cells are "nearly identical" (subsection “Revisiting the Net-fra phenotypes”, third paragraph). However, the contribution of the deepest reaching component in Figure 4 seems much more substantial than that in Figure 4, arguing that they are not nearly identical. Figure 4 appear more similar (although not in the first, most superficial component).*

The means and standard deviations of the C3 components for *fra* and Net are nearly identical. While it is true that C3 has twice the contribution to the composite distribution in Net as it does in *fra*, the salient point here is that the Net distribution is well-described without a component that is similar to wildtype – see Figure 4. This point is clarified in the revised manuscript (subsection “Re-visiting the Net-*fra* phenotypes”, third paragraph).

*A distinction is made between fra and Trim9, in that the former is absolutely required for tip expansion, but the latter is not. On this point, Net is glossed over, but Figure 6 appears to show at least one example of tip expansion in the absence of Net. If true, this may also argue for Net-independent functions of fra.*

We appreciate the reviewer’s attention to detail. The experiments presented on role of Net in R8 targeting were carried out in whole animal mutants by contrast to the cell type specific MARCM approach taken for *fra* and Trim9. In addition to *fra*, Net is a ligand for other receptors such as Unc-5 and Dscam and many cell types in addition to R8 are likely to be using this ligand during development. Indeed, we did report two additional cell types whose morphologies were altered in the whole animal Net null background. As such, we do not expect the observed R8 phenotypes to arise solely due to the absence of Net-Fra signaling in R8; pleitropy was expected and observed due to what is likely to be an altered and inhomogeneous medulla environment. In this context, the quantitative re-assessment of the mutant phenotypes and the epistasis analysis were critical in establishing that for R8 targeting removing either the ligand or the receptor were equivalent perturbations. These results justified our focus on the component with the cleaner genetics, *fra*, when identifying patterns of cell behavior and drawing mechanistic conclusions. The mutant phenotypes in the Net null background are broadly similar but there are more exceptions to each pattern. The example the reviewer noted is one such exception. As we note in the text, we do see transient dilations and other elaborations at the R8 tip in *fra* mutants as well; however, these differ from wildtype tip expansion in that they do not last. To us, the features noted by the reviewer look more like these transient dynamics than true tip expansion. The salient point is that, given the genetic background, it is not possible to ascribe such specific morphologies solely to a cell autonomous effect of the Net null mutation on R8 development. We did try a number of other approaches for a more ‘surgical’ removal of Net from the distal medulla; these are detailed in the Discussion. Unfortunately, these alternatives were not free from their own technical difficulties and caveats.